# A Near-Optimal Algorithm for Stochastic Bilevel Optimization via Double-Momentum

**Prashant Khanduri**
University of Minnesota
khand095@umn.edu

**Siliang Zeng**
University of Minnesota
zeng0176@umn.edu

**Mingyi Hong**[*]
University of Minnesota
mhong@umn.edu

**Hoi-To Wai**
CUHK
htwai@se.cuhk.edu.hk

**Zhaoran Wang**
Northwestern University
zhaoranwang@gmail.com

**Zhuoran Yang**
Princeton University
zy6@princeton.edu

## Abstract

This work proposes a new algorithm – the Single-timescale Double-momentum Stochastic Approximation (SUSTAIN) – for tackling stochastic unconstrained bilevel optimization problems. We focus on bilevel problems where the lower level subproblem is strongly-convex and the upper level objective function is smooth. Unlike prior works which rely on *two-timescale* or *double loop* techniques, we design a stochastic momentum-assisted gradient estimator for both the upper and lower level updates. The latter allows us to control the error in the stochastic gradient updates due to inaccurate solution to both subproblems. If the upper objective function is smooth but possibly non-convex, we show that SUSTAIN requires $\mathcal{O}(\epsilon^{-3/2})$ iterations (each using $\mathcal{O}(1)$ samples) to find an $\epsilon$-stationary solution. The $\epsilon$-stationary solution is defined as the point whose squared norm of the gradient of the outer function is less than or equal to $\epsilon$. The total number of stochastic gradient samples required for the upper and lower level objective functions match the best-known complexity for single-level stochastic gradient algorithms. We also analyze the case when the upper level objective function is strongly-convex.

## 1 Introduction

Many learning and inference problems take a "hierarchical" form, wherein the optimal solution of one problem affects the objective function of others [27]. Bilevel optimization is often used to model problems of this kind with two levels of hierarchy [27, 8], where the variables of an *upper level* problem depend on the optimizer of certain *lower level* problem. In this work, we consider unconstrained bilevel optimization problems of the form:

$$
\begin{aligned}
\min_{x \in \mathbb{R}^{d_{\mathsf{up}}}} \ \ell(x) &= f(x, y^*(x)) \coloneqq \mathbb{E}_\xi[f(x, y^*(x); \xi)] \\
\text{s.t.} \quad y^*(x) &= \arg\min_{y \in \mathbb{R}^{d_{\mathsf{lo}}}} \left\{ g(x, y) \coloneqq \mathbb{E}_\zeta[g(x, y; \zeta)] \right\},
\end{aligned}
\tag{1}
$$

where $f, g : \mathbb{R}^{d_{\mathsf{up}}} \times \mathbb{R}^{d_{\mathsf{lo}}} \to \mathbb{R}$ with $x \in \mathbb{R}^{d_{\mathsf{up}}}$ and $y \in \mathbb{R}^{d_{\mathsf{lo}}}$; $f(x, y; \xi)$ with $\xi \sim \pi_f$ (resp. $g(x, y; \zeta)$ with $\zeta \sim \pi_g$) represents a stochastic sample of the upper level objective (resp. lower level objective). Note here that the *upper level objective $f$* depends on the minimizer of the *lower level objective $g$*, and we refer to $\ell(x)$ as the *outer function*. Throughout this paper, $g(x, y)$ is assumed to be strongly-convex in $y$, which implies that $\ell(x)$ is smooth but possibly non-convex.

The applications of (1) include many machine learning problems that have a hierarchical structure. Examples are meta learning [13, 31], data hyper-cleaning [35], hyper-parameter optimization [12,

---

[*]Corresponding Author: Mingyi Hong.

35th Conference on Neural Information Processing Systems (NeurIPS 2021).

| Algorithm | Sample (Upper, Lower) | Implementation | Batch Size | Per-Iteration Complexity |
|---|---|---|---|---|
| BSA [14] | $\mathcal{O}(\epsilon^{-2})$, $\mathcal{O}(\epsilon^{-3})$ | Double loop | $\mathcal{O}(1)$ | $\mathcal{O}(d_{\mathsf{lo}}^2 \cdot \log T)$ |
| stocBiO [19] | $\mathcal{O}(\epsilon^{-2})$, $\mathcal{O}(\epsilon^{-2})$ | Double loop | $\mathcal{O}(\epsilon^{-1})$ | $\mathcal{O}(d_{\mathsf{lo}}^2 \cdot \log T)$ |
| TTSA [18] | $\mathcal{O}(\epsilon^{-5/2})$, $\mathcal{O}(\epsilon^{-5/2})$ | Single loop | $\mathcal{O}(1)$ | $\mathcal{O}(d_{\mathsf{lo}}^2 \cdot \log T)$ |
| STABLE [5] | $\mathcal{O}(\epsilon^{-2})$, $\mathcal{O}(\epsilon^{-2})$ | Single loop | $\mathcal{O}(1)$ | $\mathcal{O}(d_{\mathsf{lo}}^3)$ |
| SVRB [17] | $\mathcal{O}(\epsilon^{-3/2})$, $\mathcal{O}(\epsilon^{-3/2})$ | Single loop | $\mathcal{O}(1)$ | $\mathcal{O}(d_{\mathsf{lo}}^3)$ |
| SUSTAIN (this work) | $\mathcal{O}(\epsilon^{-3/2})$, $\mathcal{O}(\epsilon^{-3/2})$ | Single loop | $\mathcal{O}(1)$ | $\mathcal{O}(d_{\mathsf{lo}}^2 \cdot \log T)$ |

Table 1: Comparison of the number of upper and lower level gradient samples required to achieve an $\epsilon$-stationary point in Definition 1.1. For the algorithms with $\mathcal{O}(d_{\mathsf{lo}}^2 \cdot \log T)$ per-iteration dependence, the Hessian inverse can be computed via matrix vector products; algorithms with $\mathcal{O}(d_{\mathsf{lo}}^3)$ dependency requires Hessian inverses and Hessian projections, which incur heavy computational cost.

13, 29], and reinforcement learning [22], etc.. To better contextualize our study, below we describe examples on meta-learning problem and data hyper-cleaning problem:

*Example 1: Meta learning.* The meta learning problem aims to learn task specific parameters that generalize to a diverse set of tasks [30]. Suppose we have $M$ tasks $\{\mathcal{T}_i, i = 1, \ldots, M\}$ and each task has a corresponding loss function $L(x, y_i; \xi_i)$ with $\xi_i$ representing a data sample for task $\mathcal{T}_i$, $x \in \mathbb{R}^{d_{\mathsf{up}}}$ the model parameters shared among tasks, and $y_i \in \mathbb{R}^{d_{\mathsf{lo}}^i}$ the task specific parameters. The goal of meta learning is then to solve the following problem:

$$\min_{x \in \mathbb{R}^{d_{\mathsf{up}}}} \left\{ L_{\mathsf{ts}}(x, \bar{y}^*(x)) := \frac{1}{M} \sum_{i=1}^M \mathbb{E}_{\xi_i \sim \mathcal{D}_i}[L(x, y_i^*(x); \xi_i)] \right\}$$

$$\text{s.t. } \bar{y}^*(x) \in \arg\min_{\bar{y} \in \mathbb{R}^{\Sigma_{i=1}^M d_{\mathsf{lo}}^i}} L_{\mathsf{tr}}(x, \bar{y}) := \frac{1}{M} \sum_{i=1}^M \left( \mathbb{E}_{\zeta_i \sim \mathcal{S}_i}[L(x, y_i; \zeta_i)] + \mathcal{R}(y_i) \right), \quad (2)$$

where $\bar{y} = [y_1^T, \ldots, y_M^T]^T$, $\mathcal{R}(\cdot)$ is a strongly convex regularizer while $\mathcal{S}_i$ and $\mathcal{D}_i$ are the training and testing datasets for task $\mathcal{T}_i$. Compared to the number of tasks, the dataset sizes are usually small for meta-learning problems, so the stochasticity in tackling (2) results from the fact that at each iteration we can only sample a subset $m$ out of $M$ tasks. Note that this problem is a special case of (1). $\quad\square$

*Example 2: Data hyper-cleaning.* The data hyper-cleaning is a hyperparameter optimization problem that aims to train a classifier model with a dataset of randomly corrupted labels [35]. The optimization problem is formulated below:

$$\min_{x \in \mathbb{R}^{d_{\mathsf{up}}}} \ell(x) := \sum_{i \in \mathcal{D}_{\mathsf{val}}} L(a_i^\top y^*(x), b_i) \quad (3)$$

$$\text{s.t. } y^*(x) = \arg\min_{y \in \mathbb{R}^{d_{\mathsf{lo}}}} \left\{ c\|y\|^2 + \sum_{i \in \mathcal{D}_{\mathsf{tr}}} \sigma(x_i) L(a_i^\top y, b_i) \right\}.$$

In this problem, we have $d_{\mathsf{up}} = |\mathcal{D}_{\mathsf{tr}}|$ and $d_{\mathsf{lo}}$ is the dimension of the classifier. Moreover, $(a_i, b_i)$ is the $i$th data point; $L(\cdot)$ is the loss function, with $y$ being the model parameter; $x_i$ is the parameter that determines the weight for the $i$th data sample, and $\sigma : \mathbb{R} \to \mathbb{R}_+$ is the weight function; $c > 0$ is a regularization parameter; $\mathcal{D}_{\mathsf{val}}$ and $\mathcal{D}_{\mathsf{tr}}$ are validation and training sets, respectively. Clearly, (3) is a special case of (1) where the lower level problem finds the classifier $y^*(x)$ with the training set $\mathcal{D}_{\mathsf{tr}}$, and the upper level problem finds the best weights $x$ with respect to the validation set $\mathcal{D}_{\mathsf{val}}$. $\quad\square$

A natural approach to tackling (1) is to apply alternating stochastic gradient (SG) updates. Let $\beta, \alpha > 0$ be some step sizes, one performs the recursion

$$y^+ \leftarrow y - \beta \hat{\nabla}_y g(x, y), \quad x^+ \leftarrow x - \alpha \hat{\nabla}_x \hat{\ell}(x; y) \quad (4)$$

such that $\hat{\nabla}_y g(x, y)$, $\hat{\nabla}_x \hat{\ell}(x; y)$ are stochastic estimates of $\nabla_y g(x, y)$, $\nabla \ell(x)$, respectively. Notice that (4) is significantly different from the standard alternating primal-dual gradient algorithm for saddle point problems. Particularly, the design of $\hat{\nabla}_x \hat{\ell}(x; y)$ is crucial to the SG scheme in (4). Observe that $\nabla \ell(x)$ can be computed using the implicit function theorem, and its evaluation requires $f(\cdot, \cdot)$ and $y^\star(x)$, the minimizer of $g(x, y)$ given $x$ (cf. (5)). This gives rise to a unique challenge to bilevel optimization, where $y^\star(x)$ can only be *approximated* by $y$ obtained in the first relation of (4).

In light of the above observations, previous endeavors have considered two approaches to improve the estimate of $y^\star(x)$ while $\hat{\nabla}_x \hat{\ell}(x; y)$ is used as a *biased* approximation of $\nabla \ell(x)$. The first approach is to apply the *double-loop* algorithms. For example, [14] proposed to repeat the $y^+$ update for multiple times to obtain a better estimate of $y^\star(x)$ before performing the $x^+$ update, [19] proposed

to take a large batch size to estimate $\nabla_y g(x,y)$. While simple to analyze, these algorithms may suffer from a poor sample complexity for the inner problem. The second approach is to apply *single-loop* algorithms where the $y^+$-updates are performed simultaneously with the $x^+$-updates. Instead, advanced techniques are utilized that allows $y^+$ to accurately track $y^\star(x)$. For example, [18] suggested to tune the step size schedule with $\beta \gg \alpha$, [5, 17] proposed single-timescale algorithms with advanced variance reduction techniques. However, the latter two algorithms require Hessian projections onto a compact set along with Hessian matrices inversion which scales poorly with dimension (i.e., in $\mathcal{O}(d_{\mathsf{lo}}^3)$). We summarize and compare the complexity results of the state-of-the-art algorithms in Table 1.

A careful inspection on the above results reveals a gap in the iteration/sample complexity compared to *single-level* stochastic optimization. For instance, an optimal stochastic gradient algorithm finds an $\epsilon$-stationary solution [cf. Definition 1.1] to $\min_x \mathbb{E}_\xi[\ell(x;\xi)]$ in $\mathcal{O}(\epsilon^{-3/2})$ iterations [10, 7, 37, 42]. For bilevel optimization, the fastest rate available is only $\mathcal{O}(\epsilon^{-2})$ to the best of the authors' knowledge. In comparison, the proposed algorithm achieves a rate of $\mathcal{O}(\epsilon^{-3/2})$. During the preparation of the current paper, a preprint [17] has appeared which extended [5], and achieves an improved rate of $\mathcal{O}(\epsilon^{-3/2})$. We remark that the latter work follows a different design philosophy from ours and maybe less efficient; see the detailed discussion at the end of Sec. 3.

**Contributions.**   In this paper, we depart from the prior developments which focused on finding better inner solutions $y^*(x)$ to approximate $\hat{\nabla}_x \hat{\ell}(x;y) \approx \nabla \ell(x)$. Our idea is to exploit the gradient estimates from prior iterations to improve the quality of the current gradient estimation. This leads to *momentum*-assisted stochastic gradient estimators for *both* $\nabla_y g(x,y)$ and $\nabla \ell(x)$ using similar techniques in [7, 37] for single-level stochastic optimization. The resultant algorithm only requires $O(1)$ samples at each update, and updates $x$ and $y$ using step sizes of the same order, hence the name single-timescale double-momentum stochastic approximation(SUSTAIN) algorithm. Additionally, it is worth noting that our algorithm has a $\mathcal{O}(d_{\mathsf{lo}}^2)$ per iteration complexity, compared to the $\mathcal{O}(d_{\mathsf{lo}}^3)$ complexity of STABLE [5] and SVRB [17]. That is, the SUSTAIN algorithm is both *sample and computation* efficient. Our specific contributions are:

- We propose the SUSTAIN algorithm for bilevel problems which matches the best complexity bounds as the optimal SGD algorithms for single-level stochastic optimization. That is, it requires $\mathcal{O}(\epsilon^{-3/2})$ [resp. $\mathcal{O}(\epsilon^{-1})$] samples to find an $\epsilon$-stationary solution for non-convex (resp. strongly-convex) bilevel problems; see Table 1. Furthermore, the algorithm utilizes a single-loop update with step sizes of the same order for both upper and lower level problems. Such complexity bounds match the optimal sample complexity of stochastic gradient algorithms for single-level problems.

- By developing the Lipschitz continuous property of the (biased) stochastic estimates of $\nabla \ell(x)$, we show that obtaining a good estimate of $\nabla \ell(x)$ does not require explicit (sampled) Hessian inversion. This key result ensures that our algorithm depends favorably on the problem dimension.

- Comparing with prior works such as TTSA [18], BSA [14], STABLE [5] and SVRB [17], our analysis reveals that improving the gradient estimation quality for *both* $\nabla_y g(x,y)$ and $\nabla \ell(x)$ is the key to obtain a sample and computation efficient stochastic algorithm for bilevel optimization.

**Related works.**   The study of the bilevel problem (1) can be traced to that of game theory [36] and was formally introduced in [2–4]. It is also related to the broader class of problems of Mathematical Programming with Equilibrium Constraints [26]. Related algorithms include approximate descent [9, 38], and penalty-based methods [40]; see [6] and [25] for a comprehensive survey.

In addition to the works cited in Table 1, recent works on bilevel optimization have focused on algorithms with provable convergence rates. In [34], the authors proposed BigSAM algorithm for solving simple bilevel problems (with a single variable) with convex lower level problem. Subsequently, the works [24, 23] utilized BigSAM and developed algorithms for a general bilevel problem for the cases when the solution of the lower level problem is not a singleton. Note that all the aforementioned works [34, 24, 23] assumed the upper level problem to be strongly-convex with convex lower level problem. In a separate line of work, backpropagation based algorithms have been proposed to approximately solve bilevel problems [12, 35, 16, 15]. However, the major focus of these works was to develop efficient gradient estimators rather than on developing efficient optimization algorithms.

**Notation.**   For any $x \in \mathbb{R}^d$, we denote $\|x\|$ as the standard Euclidean norm; as for $X \in \mathbb{R}^{n \times d}$, $\|X\|$ is induced by the Euclidean norm. For a multivariate function $f(x,y)$, the notation $\nabla_x f(x,y)$ [resp. $\nabla_y f(x,y)$] refers to the partial gradient taken with respect to (w.r.t.) $x$ [resp. $y$]. For some $\mu >$

0, a function $f(x, y)$ is said to be $\mu$-strongly-convex in $x$ if $f(x, y) - \frac{\mu}{2}\|x\|^2$ is convex in $x$. For some $L > 0$, the map $\mathcal{A} : \mathbb{R}^d \to \mathbb{R}^m$ is said to be $L$-Lipschitz continuous if $\|\mathcal{A}(x) - \mathcal{A}(y)\| \le L\|x - y\|$ for any $x, y \in \mathbb{R}^d$. A function $f : \mathbb{R}^d \to \mathbb{R}$ is said to be $L$-smooth if its gradient is $L$-Lipschitz continuous. Uniform distribution over a discrete set $\{1, \ldots, T\}$ is represented by $\mathcal{U}\{1, \ldots, T\}$.

Finally, we state the following definitions for the optimality criteria of (1).

**Definition 1.1** ($\epsilon$-Stationary Point). A point $x$ is called $\epsilon$-stationary if $\|\nabla\ell(x)\|^2 \le \epsilon$. A stochastic algorithm is said to achieve an $\epsilon$-stationary point in $t$ iterations if $\mathbb{E}[\|\nabla\ell(x_t)\|^2] \le \epsilon$, where the expectation is over the stochasticity of the algorithm until time instant $t$.

**Definition 1.2** ($\epsilon$-Optimal Point). A point $x$ is called $\epsilon$-optimal if $\ell(x) - \ell^* \le \epsilon$, where $\ell^* := \min_{x \in \mathbb{R}^{d_{\mathsf{up}}}} \ell(x)$. A stochastic algorithm is said to achieve an $\epsilon$-optimal point in $t$ iterations if $\mathbb{E}[\ell(x_t) - \ell^*] \le \epsilon$, where the expectation is over the stochasticity of the algorithm until time instant $t$.

## 2 Preliminaries

We discuss the assumptions on (1) to specify the problem class of interest. We also preface the proposed algorithm by describing a practical procedure for estimating the stochastic gradients.

**Assumption 1** (Upper Level Function). $f(x, y)$ satisfies the following conditions:

(i) $\nabla_x f(x, y)$ and $\nabla_y f(x, y)$ are Lipschitz continuous w.r.t. $(x, y) \in \mathbb{R}^{d_{\mathsf{up}}} \times \mathbb{R}^{d_{\mathsf{lo}}}$, and with constants $L_{f_x} \ge 0$ and $L_{f_y} \ge 0$, respectively.

(ii) For any $(x, y) \in \mathbb{R}^{d_{\mathsf{up}}} \times \mathbb{R}^{d_{\mathsf{lo}}}$, we have $\|\nabla_y f(x, y)\| \le C_{f_y}$, for some $C_{f_y} \ge 0$.

**Assumption 2** (Lower level Function). $g(x, y)$ satisfies the following conditions:

(i) For any $x \in \mathbb{R}^{d_{\mathsf{up}}}$ and $y \in \mathbb{R}^{d_{\mathsf{lo}}}$, $g(x, y)$ is twice continuously differentiable in $(x, y)$.

(ii) $\nabla_y g(x, y)$ is Lipschitz continuous w.r.t. $(x, y) \in \mathbb{R}^{d_{\mathsf{up}}} \times \mathbb{R}^{d_{\mathsf{lo}}}$, and with constant $L_g \ge 0$.

(iii) For any $x \in \mathbb{R}^{d_{\mathsf{up}}}$, $g(x, \cdot)$ is $\mu_g$-strongly-convex in $y$ for some $\mu_g > 0$.

(iv) $\nabla^2_{xy} g(x, y)$ and $\nabla^2_{yy} g(x, y)$ are Lipschitz continuous w.r.t. $(x, y) \in \mathbb{R}^{d_{\mathsf{up}}} \times \mathbb{R}^{d_{\mathsf{lo}}}$, and with constants $L_{g_{xy}} \ge 0$ and $L_{g_{yy}} \ge 0$, respectively.

(v) For any $(x, y) \in \mathbb{R}^{d_{\mathsf{up}}} \times \mathbb{R}^{d_{\mathsf{lo}}}$, we have $\|\nabla^2_{xy} g(x, y)\|^2 \le C_{g_{xy}}$ for some $C_{g_{xy}} > 0$.

**Assumption 3** (Stochastic Functions). Assumptions 1 and 2 hold for $f(x, y; \xi)$ and $g(x, y; \zeta)$, for all $\xi \in \mathrm{supp}(\pi_f)$ and $\zeta \in \mathrm{supp}(\pi_g)$ where $\mathrm{supp}(\pi)$ is the support of $\pi$. Moreover, we assume the following variance bounds.

$$\mathbb{E}\big[\|\nabla_x f(x, y) - \nabla_x f(x, y; \xi)\|^2\big] \le \sigma^2_{f_x}, \quad \mathbb{E}\|\nabla_y f(x, y) - \nabla_y f(x, y; \xi)\|^2 \le \sigma^2_{f_y},$$

$$\mathbb{E}\|\nabla^2_{xy} g(x, y) - \nabla^2_{xy} g(x, y; \xi)\|^2 \le \sigma^2_{g_{xy}} \text{ for some } \sigma_{f_x} \ge 0, \sigma_{f_y} \ge 0 \text{ and } \sigma_{g_{xy}} \ge 0.$$

These assumptions are standard in the analysis of bilevel optimization [14]. For example, they are satisfied by a range of applications such as the meta learning problem (2), data hypercleaning problem (3) with linear classifier. Notice that under these assumptions, the gradient $\nabla\ell(\cdot)$ is well-defined. By utilizing Assumption 2–(i) and (ii) along with the implicit function theorem [33], it is easy to show that for a given $\bar{x} \in \mathbb{R}^{d_{\mathsf{up}}}$, the following holds [14, Lemma 2.1]:

$$\nabla\ell(\bar{x}) = \nabla_x f(\bar{x}, y^*(\bar{x})) - \nabla^2_{xy} g(\bar{x}, y^*(\bar{x}))[\nabla^2_{yy} g(\bar{x}, y^*(\bar{x}))]^{-1} \nabla_y f(\bar{x}, y^*(\bar{x})). \tag{5}$$

Obtaining $y^*(x)$ in closed-form is usually a challenging task, so it is natural to use the following gradient surrogate. At any $(\bar{x}, \bar{y}) \in \mathbb{R}^{d_{\mathsf{up}} \times d_{\mathsf{lo}}}$, define:

$$\bar{\nabla} f(\bar{x}, \bar{y}) = \nabla_x f(\bar{x}, \bar{y}) - \nabla^2_{xy} g(\bar{x}, \bar{y})[\nabla^2_{yy} g(\bar{x}, \bar{y})]^{-1} \nabla_y f(\bar{x}, \bar{y}). \tag{6}$$

Evaluating (6) requires computing the exact gradients and Hessian inverse which can be non-trivial. Below, we describe a practical procedure from [14] to generate a *biased* estimate of $\bar{\nabla} f(\bar{x}, \bar{y})$.

**Stochastic gradient estimator for $\nabla\ell(x)$.** The estimator requires a parameter $K \in \mathbb{N}$ and is based on a collection of $K + 3$ independent samples $\bar{\xi} := \{\xi, \zeta^{(0)}, \ldots, \zeta^{(K)}, \mathsf{k}(K)\}$, where $\xi \sim \mu$, $\zeta^{(i)} \sim \pi_g$, $i = 0, \ldots, K$, and $\mathsf{k}(K) \sim \mathcal{U}\{0, \ldots, K - 1\}$. We set

$$\bar{\nabla} f(x, y; \bar{\xi}) = \nabla_x f(x, y; \xi) - \frac{K}{L_g} \nabla^2_{xy} g(x, y; \zeta^{(0)}) \prod_{i=1}^{\mathsf{k}(K)} \left(I - \frac{\nabla^2_{yy} g(x, y; \zeta^{(i)})}{L_g}\right) \nabla_y f(x, y; \xi), \tag{7}$$

where we have used the convention $\prod_{i=1}^{j} A_i = I$ if $j = 0$. It has been shown in [14, 18] that the bias with the gradient estimator (7) decays exponentially fast with $K$, as summarized below:

**Lemma 2.1.** *Under Assumptions 1, 2. For any $K \geq 1$, the gradient estimator in (7) satisfies*

$$\|\bar{\nabla} f(x, y) - \mathbb{E}_{\bar{\xi}}[\bar{\nabla} f(x, y; \bar{\xi})]\| \leq \frac{C_{g_{xy}} C_{f_y}}{\mu_g} \left( 1 - \frac{\mu_g}{L_g} \right)^K, \quad \forall (x, y) \in \mathbb{R}^{d_{\mathsf{up}}} \times \mathbb{R}^{d_{\mathsf{lo}}}. \tag{8}$$

The detailed statement of the above lemma is included in Appendix C. We remark that each computation of $\bar{\nabla} f(x, y; \bar{\xi})$ requires at most $K$ Hessian-vector products, and later we will show that setting $K = \mathcal{O}(\log(T))$ is necessary for the proposed algorithm. Since $\nabla^2_{yy} g(x, y; \zeta)$ is of size $d_{\mathsf{lo}} \times d_{\mathsf{lo}}$, the total complexity of this step is $\mathcal{O}(\log(T) d_{\mathsf{lo}}^2)$. On the contrary, STABLE [5] and SVRB [17] require $\mathcal{O}(d_{\mathsf{lo}}^3)$ to estimate the Hessian inverse, which is more computationally expensive when $d_{\mathsf{lo}} \gg 1$. Indeed, it has been explicitly mentioned in [5] that "*our algorithm (STABLE) is preferable in the regime where the sampling is more costly than computation or the dimension $d$ is relatively small*".

Notice that (7) is not the only option for estimating the gradient surrogate $\bar{\nabla} f(x, y)$. For ease of presentation, below we abstract out the conditions on the stochastic estimates of $\nabla_y g, \bar{\nabla} f$ required by our analysis as the following assumption:

**Assumption 4** (Stochastic Gradients). *For any $(x, y) \in \mathbb{R}^{d_{\mathsf{up}}} \times \mathbb{R}^{d_{\mathsf{lo}}}$, there exists constants $\sigma_f, \sigma_g \geq 0$ such that the estimates $\nabla_y g(x, y; \zeta), \bar{\nabla} f(x, y; \bar{\xi})$ satisfy:*

(i) The gradient estimate of the upper level objective satisfies:

$$\mathbb{E}_{\bar{\xi}} \left[ \|\bar{\nabla} f(x, y; \bar{\xi}) - \bar{\nabla} f(x, y) - B(x, y)\|^2 \right] \leq \sigma_f^2, \tag{9}$$

where $B(x, y) = \mathbb{E}_{\bar{\xi}}[\bar{\nabla} f(x, y; \bar{\xi})] - \bar{\nabla} f(x, y)$ is the bias in estimating $\bar{\nabla} f(x, y)$.

(ii) The gradient estimate of the lower level objective satisfies

$$\mathbb{E}_{\zeta} \left[ \|\nabla_y g(x, y; \zeta) - \nabla_y g(x, y)\|^2 \right] \leq \sigma_g^2. \tag{10}$$

As observed from Lemma 2.1, the gradient estimator (7) satisfies Assumption 4(i).

Lastly, the approximate gradient defined in (6), the true gradient (5), as well as the optimal solution of the lower level problem are Lipschitz continuous, as proven below:

**Lemma 2.2.** *[14, Lemma 2.2] Under Assumptions 1, 2 and 3, we have*

$$\|\bar{\nabla} f(x, y) - \nabla \ell(x)\| \leq L\|y^*(x) - y\|, \quad \|y^*(x_1) - y^*(x_2)\| \leq L_y \|x_1 - x_2\|,$$
$$\|\nabla \ell(x_1) - \nabla \ell(x_2)\| \leq L_f \|x_1 - x_2\|, \tag{11}$$

*for all $x, x_1, x_2 \in \mathbb{R}^{d_{\mathsf{up}}}$ and $y \in \mathbb{R}^{d_{\mathsf{lo}}}$. The above Lipschitz constants are defined as:*

$$L = L_{f_x} + \frac{L_{f_y} C_{g_{xy}}}{\mu_g} + C_{f_y} \left( \frac{L_{g_{xy}}}{\mu_g} + \frac{L_{g_{yy}} C_{g_{xy}}}{\mu_g^2} \right), \quad L_f = L + \frac{L C_{g_{xy}}}{\mu_g}, \quad L_y = \frac{C_{g_{xy}}}{\mu_g}. \tag{12}$$

The first result in (11) reveals that $\bar{\nabla} f(x, y)$ approximates $\nabla \ell(x)$ when $y \approx y^*(x)$. This suggests that a *double-loop* algorithm which solves the strongly-convex lower level problem to sufficient accuracy can be applied to tackle (1). Such approach has been pursued in [14, 19]. Next, we propose an algorithm which rely on *single-loop* updates with improved sample efficiency.

## 3 The proposed SUSTAIN algorithm

Equipped with a practical stochastic gradient estimator for $\nabla \ell(x)$ [cf. (7)], our next endeavor is to develop a *single-loop* algorithm to tackle (1) through drawing $\mathcal{O}(1)$ samples for upper and lower level problems at each iteration. Our main idea is to adopt the recursive momentum techniques developed in [7, 37]. Notice that these works utilize *unbiased* stochastic gradients evaluated at consecutive iterates to construct a variance reduced gradient estimate for single-level stochastic optimization.

In the context of bilevel stochastic optimization (1), a few key challenges are in order:

---
**Algorithm 1** The Proposed SUSTAIN Algorithm
---
1: **Input**: Parameters: $\{\beta_t\}_{t=0}^{T-1}$, $\{\alpha_t\}_{t=0}^{T-1}$, $\{\eta_t^f\}_{t=0}^{T-1}$, and $\{\eta_t^g\}_{t=0}^{T-1}$ with $\eta_0^f = \eta_0^g = 1$
2: **Initialize**: $x_0$, $y_0$; set $x_{-1} = y_{-1} = h_{-1}^f = h_{-1}^g = 0$
3: **for** $t = 0$ to $T-1$ **do**
4:    ($y$-update) Compute the gradient estimator $h_t^g$ by (13) and set $y_{t+1} = y_t - \beta_t h_t^g$.
5:    ($x$-update) Compute the gradient estimator $h_t^f$ by (14) and set $x_{t+1} = x_t - \alpha_t h_t^f$.
6: **end for**
7: **Return:** $x_{a(T)}$ where $a(T) \sim \mathcal{U}\{1, ..., T\}$.
---

- Recall from Lemma 2.1 that obtaining an unbiased estimator for the outer gradient $\nabla\ell(x)$ requires using $K \to \infty$ samples in (7), this calls for the new techniques to control the bias arising from approximating $\nabla\ell(x)$.

- The gradient estimator (7) has a more complicated structure than a plain gradient estimator, as it involves up to three different stochastic vectors/matrices related to $\nabla_x f(x,y)$, $\nabla_y f(x,y)$, $\nabla_{xy} g(x,y)$, and one stochastic inversion that is related to $[\nabla_{yy} g(x,y)]^{-1}$. It is not clear which are the most important objects for which variance reduction shall be applied.

Our key innovation is to develop a useful estimate of $\bar\nabla f(x,y)$ by using a novel *double-momentum* technique. First, we build a recursive momentum estimator for $\nabla_y g(x,y)$, based upon which the variable $y$ gets updated. Then, with such a "stabilized" inner iteration, we compute an estimate of $\bar\nabla f(x,y)$ as given in (7), by using the four stochastic vectors/matrices mentioned above but without performing any variance reduction. Such a stochastic estimator will then be used to construct a recursive momentum estimator for $\bar\nabla f(x,y)$. The intuition is that as long as $y$ is *accurate enough*, then the stochastic terms in (7) are also accurate enough, so they can be used to construct the estimator for the outer gradient. Our approach only tracks two vector estimators, while still being able to leverage the low-complexity sample-based Hessian inversion as given in (7).

The SUSTAIN algorithm is summarized in Algorithm 1. Define $\eta_t^g \in [0,1]$, $\eta_t^f \in [0,1]$. For the lower level problem involving $y$, it utilizes the following momentum-assisted gradient estimator, $h_t^g \in \mathbb{R}^{d_{\text{lo}}}$, defined recursively as

$$h_t^g = \eta_t^g \nabla_y g(x_t, y_t; \zeta_t) + (1 - \eta_t^g)\big(h_{t-1}^g + \nabla_y g(x_t, y_t; \zeta_t) - \nabla_y g(x_{t-1}, y_{t-1}; \zeta_t)\big); \quad (13)$$

For the upper level problem involving $x$, we utilize a similar estimate, $h_t^f \in \mathbb{R}^{d_{\text{up}}}$, defined as

$$h_t^f = \eta_t^f \bar\nabla f(x_t, y_t; \bar\xi_t) + (1 - \eta_t^f)\big(h_{t-1}^f + \bar\nabla f(x_t, y_t; \bar\xi_t) - \bar\nabla f(x_{t-1}, y_{t-1}; \bar\xi_t)\big). \quad (14)$$

The gradient estimators $h_t^g$ and $h_t^f$ are computed from the current and past gradient estimates $\nabla_y g(x_t, y_t; \zeta_t)$, $\nabla_y g(x_{t-1}, y_{t-1}; \zeta_t)$ and $\bar\nabla f(x_t, y_t; \bar\xi_t)$, $\bar\nabla f(x_{t-1}, y_{t-1}; \bar\xi_t)$. Note that the stochastic gradients at two consecutive iterates are computed using the same sample sets $\zeta_t$ for $h_t^g$ and $\bar\xi_t$ for $h_t^f$.

Both $x$ and $y$-update steps mark a major departure of the SUSTAIN algorithm from existing algorithms on bilevel optimization [14, 18, 19]. The latter works apply the direct gradient estimator $\bar\nabla f(x_t, y_{t+1}; \bar\xi_t)$ [cf. (7)] to serve as an estimate to $\bar\nabla f(x,y)$ [and subsequently $\nabla\ell(x)$]. To guarantee convergence, these works focused on improving the *tracking performance* of $y_{t+1} \approx y^\star(x_t)$ by employing double-loop updates, e.g., by repeatedly applying SG step multiple times for the inner problem; or a sophisticated two-timescale design for the step sizes, e.g., by setting $\beta_t/\alpha_t \to \infty$.

A recent preprint [17] suggested the SVRB algorithm which applies a similar recursive momentum technique as SUSTAIN. However, SVRB is different from SUSTAIN as the momentum estimator is applied exhaustively to *all* the individual random quantities involved in (7) and requires Hessian projection. As a result, the SVRB algorithm entails a high complexity in storage and computation as the latter has to store matrix variables of size $d_{\text{lo}} \times d_{\text{lo}}$ and computes a matrix inverse for each iteration. In comparison, the SUSTAIN algorithm only requires storing the gradient estimators $h_t^g, h_t^f$ of size $d_{\text{lo}}, d_{\text{up}}$, respectively, and the computation complexity is only $\mathcal{O}(d_{\text{lo}}^2 K)$ for each iteration.

## 3.1 Convergence analysis

In the following, we present the convergence analysis for the SUSTAIN algorithm when $\ell(\cdot)$ is a smooth function [cf. consequence of Assumptions 1, 2 and 3]. Before proceeding to the main results, we present a lemma about the Lipschitzness of the gradient estimate $\bar{\nabla} f(x, y; \bar{\xi})$ given in (7):

**Lemma 3.1.** *Under Assumptions 1, 2 and 3, we have for any* $(x_1, y_1), (x_2, y_2) \in \mathbb{R}^{d_{up}} \times \mathbb{R}^{d_{lo}}$,

$$\mathbb{E}_{\bar{\xi}} \| \bar{\nabla} f(x_1, y_1; \bar{\xi}) - \bar{\nabla} f(x_2, y_2; \bar{\xi}) \| \leq L_K^2 \{ \|x_1 - x_2\| + \|y_1 - y_2\| \}^2,$$

*where*

$$L_K = \sqrt{2L_{f_x}^2 + \frac{6C_{g_{xy}}^2 L_{f_y}^2 K}{2\mu_g L_g - \mu_g^2} + \frac{6C_{f_y}^2 L_{g_{xy}}^2 K}{2\mu_g L_g - \mu_g^2} + \frac{6C_{g_{xy}}^2 C_{f_y}^2 L_{g_{yy}}^2 K^3}{(L_g - \mu_g)^2 (2\mu_g L_g - \mu_g^2)}}, \quad (15)$$

*and $K$ is the number of samples required to construct the stochastic gradient estimate given in* (7).

The detailed proof can be found in Appendix C. We remark that the above result is crucial for analyzing the error of the gradient estimate $h_t^f$ defined in (14). To see this, let us first define the errors of the gradient estimates for the outer and inner functions as follows

$$e_t^f := h_t^f - \bar{\nabla} f(x_t, y_t) - B_t, \quad e_t^g := h_t^g - \bar{\nabla}_y g(x_t, y_t), \quad (16)$$

where $B_t := B(x_t, y_t)$ denotes the bias. Rewriting $e_t^f$ using (14) gives the following recursion:

$$e_t^f = (1 - \eta_t^f) e_{t-1}^f + (1 - \eta_t^f) \{ \bar{\nabla} f(x_t, y_t; \bar{\xi}_t) - \bar{\nabla} f(x_{t-1}, y_{t-1}; \bar{\xi}_t)$$
$$- (\bar{\nabla} f(x_t, y_t) + B_t - \bar{\nabla} f(x_{t-1}, y_{t-1}) - B_{t-1}) \} + \eta_t^f (\bar{\nabla} f(x_t, y_t; \bar{\xi}_t) - \bar{\nabla} f(x_t, y_t) - B_t).$$

Lemma 3.1 allows us to control the variance of the second term in the above relation as $\mathcal{O}(\alpha_t^2 \|h_{t-1}^f\|^2 + \beta_t^2 \|h_{t-1}^g\|^2)$. This subsequently leads to a reduced error magnitude for $\mathbb{E}[\|e_t^f\|^2]$. Similarly, we can show a reduced error magnitude for $\mathbb{E}[\|e_t^g\|^2]$ for the inner gradient estimate.

The above discussion suggests that we can track the gradient $\nabla \ell(x)$ using only stochastic gradient estimates (7), without needing to track each component stochastic vectors/matrices. This allows us to avoid costly Hessian inversions. In contrast, [5, 17] track the individual stochastic vectors/matrices of (7), and then combine them together to yield an estimate of $\nabla \ell(x)$. This approach is unable to utilize the cheap stochastic estimates of Hessian and have to invert it directly.

Turning back to the convergence analysis of the SUSTAIN algorithm, the main idea of our analysis is to demonstrate reduction of a properly constructed potential function across iterations. For smooth (possibly non-convex) objective function, this potential function consists of a linear combination of the norms of the error terms $\mathbb{E}[\|e_t^f\|^2]$ and $\mathbb{E}[\|e_t^g\|^2]$ along with the outer objective function $\ell(x_t)$ and the inner optimality gap $\|y_t - y^*(x_t)\|^2$. We obtain:

**Theorem 3.2.** *Under Assumptions 1–4. Fix $T \geq 1$ as the maximum iteration number. Set the number of samples used for the gradient estimator in* (7) *as $K = (L_g/\mu_g) \log (C_{g_{xy}} C_{f_y} T/\mu_g)$ and*

$$\alpha_t = \frac{1}{(w + t)^{1/3}}, \quad \beta_t = c_\beta \alpha_t, \quad \eta_t^f = c_{\eta_f} \alpha_t^2, \quad \eta_t^g = c_{\eta_g} \alpha_t^2, \quad (17)$$

*where $w, c_\beta, c_{\eta_f}, c_{\eta_g}$ are defined in* (29) *of appendix. The iterates generated by Algorithm 1 satisfy*

$$\mathbb{E} \| \nabla \ell(x_{a(T)}) \|^2 = \mathcal{O} \left( \frac{\ell(x_0) - \ell^*}{T^{2/3}} + \frac{\|y_0 - y^*(x_0)\|^2}{T^{2/3}} + \frac{\log(T) \sigma_f^2}{T^{2/3}} + \frac{\log(T) \sigma_g^2}{T^{2/3}} \right). \quad (18)$$

Details of the constants in the theorem and its proof can be found in Appendix D. The above result shows that to reach an $\epsilon$-stationary point, the SUSTAIN algorithm requires $\widetilde{\mathcal{O}}(\epsilon^{-3/2})$ (omitting logarithmic factors) samples of stochastic gradients from both the upper and lower level functions.

This sample complexity matches the best complexity bounds for single-level stochastic optimization like SPIDER [10], STORM [7], SNVRG [42] and Hybrid SGD [37]. We claim that this is a *near-optimal* sample complexity for bilevel stochastic optimization since for example, we have imposed additional smoothness conditions on the Hessian of the lower level problem. We will leave this as an open question to investigate the lower bound complexity for bilevel stochastic optimization.

**Strongly-convex $\ell(x)$.** We also discuss the case when in addition to smoothness, $\ell(\cdot)$ is $\mu_f$-strongly-convex. Here, a stronger guarantee can be obtained:

**Theorem 3.3.** *Under Assumptions [1]–[4], and suppose $\ell(x)$ is $\mu_f$-strongly-convex. Fix any $T \geq 1$, set the number of samples for the gradient estimator ([7]) as $K = (L_g/2\mu_g) \log \left( C_{g_{xy}}^2 C_{f_y}^2 T/\mu_g^2 \right)$ and*

$$\alpha_t \equiv \alpha \leq \left\{ \frac{1}{\mu_f + 1}, \frac{1}{2\mu_g \hat{c}_\beta}, \frac{\mu_g}{\hat{c}_\beta L_g^2}, \frac{1}{8L_K^2 + L_f}, \frac{L^2 + 2L_y^2}{4L_K^2 L_g^2 \hat{c}_\beta^2} \right\}, \quad \eta_t^f \equiv (\mu_f + 1)\alpha, \quad \beta_t \equiv \hat{c}_\beta \alpha,$$

*where $\eta_t^g \equiv 1$, $\hat{c}_\beta = 8L_y^2 + 8L^2 + 2\mu_f/\mu_g$ and $L_K$ is defined in ([15]). The iterates generated by Algorithm [1] satisfy for any $t \geq 1$ that:*

$$\mathbb{E}[\ell(x_t) - \ell^*] \leq (1 - \mu_f \alpha)^t \bar{\Delta}_0 + \frac{1}{\mu_f} \left\{ \frac{2}{T} + \left[ (2\hat{c}_\beta^2 + 8\hat{c}_\beta^2 L_K^2)\sigma_g^2 + 2(\mu_f + 1)^2 \sigma_f^2 \right]\alpha \right\}, \quad (19)$$

*where $\bar{\Delta}_0 := \ell(x_0) - \ell^* + \sigma_f^2 + \|y_0 - y^*(x_0)\|^2$.*

The detailed proof can be found in Appendix [E]. For large $T$, setting $\alpha \asymp 1/T$ shows that the bound in ([19]) decreases at the rate of $\mathcal{O}(1/T)$.

Theorem [3.3] shows that to reach an $\epsilon$-optimal point, the SUSTAIN algorithm requires $\widetilde{\mathcal{O}}(\epsilon^{-1})$ stochastic gradient samples from the upper and lower level problems, also see the detailed calculations in Appendix [E]. This improves over TTSA [18] which requires $\widetilde{\mathcal{O}}(\epsilon^{-1.5})$ samples, and BSA [14] which requires $\widetilde{\mathcal{O}}(\epsilon^{-1})$, $\mathcal{O}(\epsilon^{-2})$ samples for the upper and lower level problems, respectively. Again, we achieve similar sample complexity as SGD applied on strongly-convex single-level optimization.

Interestingly, in Theorem [3.3], we have selected $\eta_t^g \equiv 1$ where the momentum term in the lower level gradient vanishes. In this way, the SUSTAIN algorithm is reduced into a *single-momentum* algorithm where the recursive momentum acceleration is only applied to the upper level gradient. Similarly, in Theorem [3.2], if SUSTAIN utilizes only the upper level momentum, i.e., $\eta_t^g \equiv 1$, then with appropriate choice of parameters, we get $\mathbb{E}\|\nabla \ell(x_{a(T)})\|^2 \leq \mathcal{O}(1/\sqrt{T})$ (please see [20] for further details). This implies that to achieve an $\epsilon$-stationary solution SUSTAIN with only upper level momentum requires $\mathcal{O}(\epsilon^{-2})$ stochastic samples for both the upper and the lower level functions. Note that this improves over TTSA [18] which utilizes a vanilla SGD update for both the upper and the lower level problems, i.e., $\eta_t^f \equiv 1$ and $\eta_t^g \equiv 1$ and requires $\mathcal{O}(\epsilon^{-5/2})$ stochastic samples for both upper and lower level functions.

# 4 Numerical experiments

In this section, we evaluate the performance of the SUSTAIN algorithm on two popular machine learning tasks: hyperparameter optimization and meta learning.

**Hyperparameter optimization.** We consider the data hyper-cleaning task ([3]), and compare SUSTAIN with several algorithms such as stocBiO [19] for different batch size choices, and the HOAG algorithm in [29]. Note that in [19], the authors shown that stocBio exhibits better practical performance compared with other bilevel optimization algorithms.

We consider problem ([3]) with $L(\cdot)$ being the cross-entropy loss (i.e., a data cleaning problem for logistic regression); $\sigma(x) := \frac{1}{1+\exp(-x)}$; $c = 0.001$; see [35]. The problem is trained on the `FashionMNIST` dataset [41] with 50k, 10k, and 10k image samples allocated for training, validation and testing purposes, respectively. The step sizes for different algorithms are chosen according to their theoretically suggested values. Let the outer iteration be indexed by $t$, for SUSTAIN we choose $\alpha_t = \beta_t = 0.1/(1 + t)^{1/3}$ and tune for $c_{\eta_f}$ and $c_{\eta_g}$ (see Theorem [3.2]), for stocBiO and HOAG we select $\alpha_t = d_\alpha$, $\beta_t = d_\beta$ and tune for parameters $d_\alpha$ and $d_\alpha$ in the range $[0, 1]$.

In Figure [1], we compare the performance of different algorithms when the dataset has a corruption probability of 0.3. As observed, SUSTAIN outperforms stocBiO and HOAG. We remark that HOAG is a deterministic algorithm and hence requires full batch gradient computations at each iteration. Similarly, stocBio relies on large batch gradients which results in relatively slow convergence. This fast convergence of SUSTAIN results form the single timescale update with reduced variance resulting from the double-momentum variance reduced updates.

**Meta learning.** We consider a few-shot meta learning problem [11, 30] (cf. ([2])) and compare the performance of SUSTAIN to ITD-BiO [19] and ANIL [30]. The task of interest is 5-way 5-shot

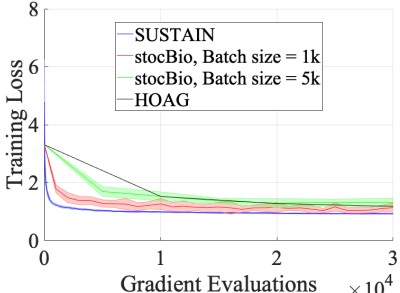
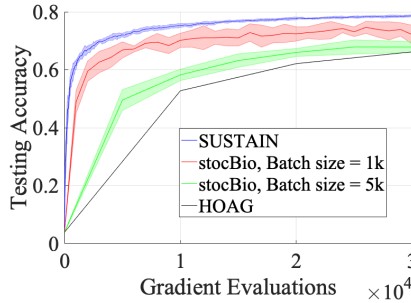

Figure 1: **Hyperparameter optimization**: Data hyper-cleaning task on the `FashionMNIST` dataset. We plot the training loss and testing accuracy against the number of gradients evaluated with corruption rate $p = 0.3$.

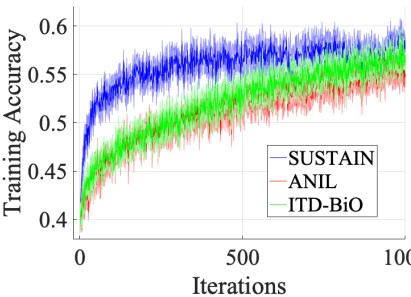
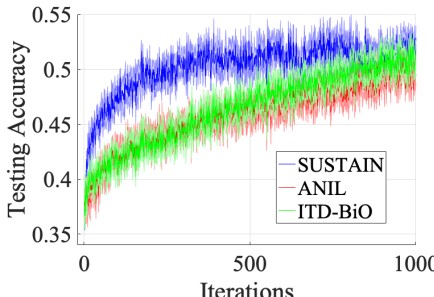

Figure 2: **Meta learning**: 5-way 5-shot learning task on the `miniImageNet` dataset. We plot the training and testing accuracy against the number of iterations.

learning and we conduct experiments on the `miniImageNet` dataset [39, 32] with 100 classes and 600 images per class. We apply `learn2learn` [1] (available: `https://github.com/learnables/learn2learn`) to partition the 100 classes from `miniImageNet` into subsets of 64, 16 and 20 for meta training, meta validation and meta testing, respectively. Similar to [1, 19], we implement a 4-layer convolutional neural network (CNN) with ReLU activation for the learning task. At each iteration, we sample a batch of 32 tasks from a set of 20000 tasks allocated for training and 600 each for validation and testing. For each algorithm, we implement 10 inner and 1 outer update. The performance is averaged over 10 Monte Carlo runs.

For ANIL and ITD-BiO, we use the parameter selection suggested in [1, 19]. Specifically, for ANIL, we use inner-loop stepsize of $0.1$ and the outer-loop (meta) stepsize as $0.002$. For ITD-BiO, we choose the inner-loop stepsize as $0.05$ and the outer-loop stepsize to be $0.005$. For SUSTAIN, we choose the outer-loop stepsize $\alpha_t$ as $\kappa/(1 + t)^{1/3}$ and choose $\kappa \in [0.1, 1]$, we choose the momentum parameter $\eta_t$ as $\bar{c}\alpha_t^2/\kappa^2$ and tune for $\bar{c} \in \{2, 5, 10, 15, 20\}$, finally, we fix the inner stepsize as $0.05$. For the outer loop update ANIL and ITD-BiO utilize SGD optimizer whereas SUSTAIN uses the hybrid gradient estimator.

From Figure 2 which compares the training and testing accuracy against the iteration number, we observe that SUSTAIN achieves a better performance compared to ANIL and ITD-BiO on the meta learning task. Also, notice that in the initial iterations SUSTAIN converges faster but then converges probably as a consequence of diminishing stepsizes (and momentum parameter). In contrast, ANIL and ITD-BiO slowly improve in performance and catch up with SUSTAIN's performance. In the appendix, we show that the SUSTAIN algorithm requires less computation time to achieve better performance compared to the ANIL and ITD-BiO.

For further evaluation of the performance of SUSTAIN, we have included additional experiments on hyperparameter optimization and meta learning on different datasets in the supplementary material.

## Conclusions and limitations

We have developed the SUSTAIN algorithm for unconstrained bilevel optimization with strongly convex lower level subproblems. The proposed algorithm executes on a single-timescale, without the need to use either two-timescale updates, large batch gradients, or double-loop algorithm. We showed that SUSTAIN  is both *sample* and *computation* efficient, because it matches the best-known sample complexity guarantees for single-level problems with non-convex and strongly convex objective (smooth) functions, while matching the best-known per-iteration computational complexity for the same class of bi-level problems. In the future, we plan to rigorously show the sample complexity lower bounds for the considered class of bilevel problems. Further, we plan to develop sample and communication efficient algorithms for a more general class of bilevel problems, such as those with constraints in the lower level problems.

## Acknowledgement

We thank the anonymous reviewers for their valuable comments and suggestions. The work of Prashant Khanduri, Siliang Zeng, and Mingyi Hong was supported by the National Science Foundation (NSF) through grant CIF-1910385. The work of Mingyi Hong was also supported by an IBM Faculty Research award. The work of Hoi-To Wai was supported by CUHK Direct Grant #4055113. Zhaoran Wang acknowledges National Science Foundation (Awards 2048075, 2008827, 2015568, 1934931), Simons Institute (Theory of Reinforcement Learning), Amazon, J.P. Morgan, and Two Sigma for their support. Zhuoran Yang acknowledges Simons Institute (Theory of Reinforcement Learning).

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
