# Appendix

## A   Additional experiments

In this section, we supplement the numerical results presented in Section 4 with additional experiments on real datasets. We demonstrate the efficacy of SUSTAIN for the meta learning and hyperparameter optimization tasks. Furthermore, we examine the performance of SUSTAIN when combined with an Adam-like update rule [cf. see Algorithm 2].

**Meta learning.**   In Figure 2 of Section 4 in the main paper, we established that when SUSTAIN, ITD-BiO and ANIL utilize vanilla SG direction for the outer level update, SUSTAIN outperforms rest of the algorithms for the meta learning problem. Specifically, we compared the training and testing performance of the algorithms with the number of iterations (i.e., the outer update $t$ in Algorithm 1). In each iteration, all the algorithms access the same number of samples while SUSTAIN requiring twice the number of gradient computations (cf. (14)). As observed from Figure 2, SUSTAIN requires the smallest number of iterations (samples) and gradient computations to achieve a given training/testing accuracy on the benchmarked dataset.

We conduct additional experiments for meta learning and demonstrate the following: (1) for the outer level update we can adapt Adam [21] optimizer with the SUSTAIN framework to achieve better performance, (2) the outer gradient estimate (14) for SUSTAIN can be designed with only one gradient computation per iteration (instead of two) without compromising performance, and (3) SUSTAIN outperforms MAML [11], ANIL [30] and ITD-BiO [19] when all algorithms implement Adam for the outer level update. Next, we discuss the datasets and the parameter settings.

We consider meta learning problem with `miniImageNet` [39, 32] and FC100 [28] datasets. Both datasets consist of 100 classes with each class containing 600 images. For the `miniImageNet`, we consider the same setting as in Section 4. For FC100, we follow the setting of [28, 19] where 100 classes are split into 60, 20 and 20 classes for meta-training, meta-validation and meta-testing, respectively. For both datasets, we consider a 5-way 5-shot learning task where the algorithm aims to classify samples into 5 unseen classes using only 5 available samples. We implement the solver using a 4-layer CNN (with different width for each dataset). We compare heuristic versions of SUSTAIN with MAML [11], ANIL [30] and recently proposed ITD-BiO [19], where these algorithms all utilize the Adam [21] solver for the outer problem's update. These heuristic algorithms are also used in [19] when comparing performance of the bilevel algorithms for meta-learning tasks. Note that these Adam-based bilevel algorithms for meta learning do not have any theoretical performance guarantees. Nevertheless, in the following we show that they perform well in practice [19].

We first discuss the parameter setting for the meta learning task using `miniImageNet` dataset. All the algorithms sample 32 tasks in each iteration. For the Adam versions of MAML, ANIL and ITD-BiO, we choose the parameters as suggested in [1, 19]. For all the algorithms, we execute 10 update steps in the inner loop followed by a single outer update step. Each update step is counted as a

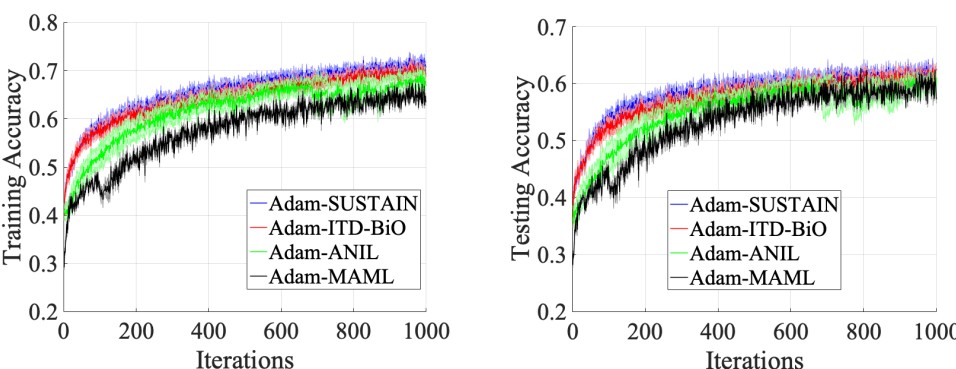

Figure 3: **Meta learning**: 5-way 5-shot learning task on the `miniImageNet` dataset. We plot the training and testing accuracy against the number of iterations with each iteration representing one outer level update step. All the algorithms utilize Adam [21] optimizer for the outer loop update.

---

**Algorithm 2** Update direction for Adam-SUSTAIN (also see footnote[1])

---

1: **Parameters**: $\gamma_1 = 0.9$, $\gamma_2 = 0.999$, $m_0 = 0$, $v_0 = 0$, $\epsilon = 10^{-8}$ and $\eta_t^f$
2: **for** $t = 1, \cdots, T$ **do**
3:     **Input**: $(x_t, y_t)$, $(x_{t-1}, y_{t-1})$ from Algorithm 1.
4:     Compute the gradient estimator $\bar{h}_t^f$ using Option I or II in (20)
5:     Update first moment estimate: $m_t \leftarrow \gamma_1 \cdot m_{t-1} + (1 - \gamma_1)\bar{h}_t^f$
6:     Bias-correction for first moment estimate: $m_t \leftarrow m_t/(1 - (\gamma_1)^t)$
7:     Update second moment estimate: $v_t \leftarrow \gamma_2 \cdot v_{t-1} + (1 - \gamma_2)(\bar{h}_t^f)^2$
8:     Bias-correction for second moment estimate: $v_t \leftarrow v_t/(1 - (\gamma_2)^t)$
9:     Use the update direction: $h_t^f \leftarrow m_t/(\sqrt{v_t} + \epsilon)$
10: **end for**
11: **Return:** $h_t^f$

---

single iteration. The implementation of MAML and ANIL is adopted from existing implementations in [1]. For MAML, we choose the inner loop stepsize to be 0.5 and the outer loop stepsize to be 0.003. For ANIL we utilize inner loop stepsize of 0.1 and outer loop stepsize of 0.002. Both ITD-BiO and SUSTAIN utilize gradient descent with stepsize of 0.05 as the inner optimizer. For the outer update ITD-BiO uses a stepsize of 0.002 (the parameters for ITD-BiO are selected based the repository https://github.com/JunjieYang97/stocBiO). For SUSTAIN we set the outer stepsize as $\alpha_t = 0.005$ and tune for the momentum parameter $\eta_t^f = \bar{c}/\kappa^2(1 + t)^{2/3}$ with fixed $\kappa = 0.005$ by choosing $\bar{c} \in \{0.25, 2.5, 5, 10\}$. In contrast to other algorithms, SUSTAIN applies Adam [21] to the hybrid stochastic gradient estimator used for the outer update (14). For detailed steps please see Algorithm 2[2]. Moreover, it is worth noting that the direction update rule Option II given in (20) is a modification of the original update given in (14) (or equivalently Option I in (20)). Such a rule requires just a single (mini-batch) gradient computation per iteration (which is the same as MAML, ANIL and ITD-BiO), and in practice, its performance is very close to that of Option I. Our results below uses Option II as the update direction.

$$\bar{h}_t^f = \begin{cases} \bar{\nabla} f(x_t, y_t; \bar{\xi}_t) + (1 - \eta_t^f)\big(\bar{h}_{t-1}^f - \bar{\nabla} f(x_{t-1}, y_{t-1}; \bar{\xi}_t)\big) & \text{Option I} \\ \bar{\nabla} f(x_t, y_t; \bar{\xi}_t) + (1 - \eta_t^f)\big(\bar{h}_{t-1}^f - \underbrace{\bar{\nabla} f(x_{t-1}, y_{t-1}; \bar{\xi}_{t-1})}_{\text{Previous SG}}\big) & \text{Option II} \end{cases} \quad (20)$$

In Figure 3, we plot the training and testing performance against the number of iterations for the Adam version SUSTAIN with other algorithms for 5-way 5-shot learning task on miniImageNet dataset. Note from the discussion above, we know that in each iteration all the algorithms access

---

[2]Note that the vector division and exponent operations in the Algorithm are implemented element wise. The values of the parameters chosen for Adam are default values used by the PyTorch library.

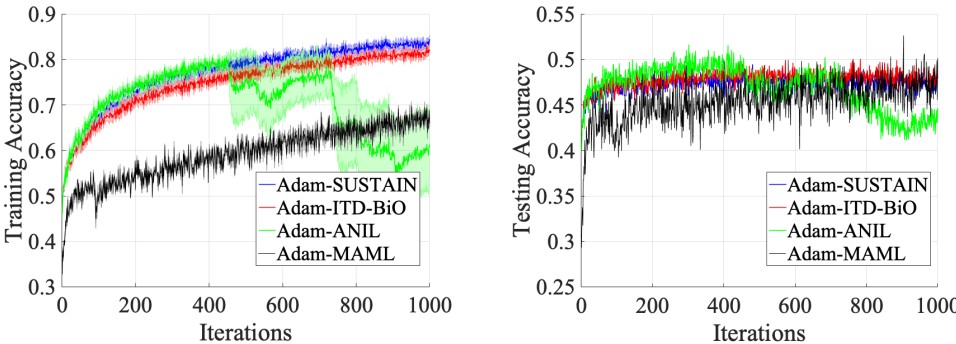

Figure 4: **Meta learning**: 5-way 5-shot learning task on the FC100 dataset. We plot the training and testing accuracy against the number of iterations with each iteration representing one outer level update. All the algorithms utilize Adam [21] optimizer for the outer loop update.

the same number of sample, and spend the same amount of (mini-batch) gradient computation efforts. Consequently, Figure 3 implies that SUSTAIN outperforms ITD-BiO, ANIL and MAML as it requires fewest iteration (thus samples and gradient computation) to achieve the improved performance. Importantly, these Adam-based algorithms significantly outperform their vanilla version (cf. Figure 2 for performance with SGD), in terms of both accuracy and speed.

Next, we compare the performance of SUSTAIN with other algorithms for the meta learning task using `FC100` dataset. For this task all the algorithms sample 32 tasks in each iteration. In contrast to the previous dataset, for this task we execute 20 update steps in the inner loop followed by a single outer update step. Similar to `miniImageNet` dataset, we adopt existing implementations of MAML and ANIL from [1] and ITD-BiO from [19]. For MAML, we choose inner loop stepsize of 0.5 and the outer loop stepsize of 0.001. For ANIL we utilize inner loop stepsize of 0.1 and outer loop stepsize of 0.001. In the inner loop, both ITD-BiO and SUSTAIN utilize gradient descent with a stepsize of 0.1. For the outer update ITD-BiO uses a stepsize of 0.001 ((the parameters for ITD-BiO are selected based the repository `https://github.com/JunjieYang97/stocBiO`)). For the outer update SUSTAIN utilizes the same setting as required for `miniImageNet` dataset and the Adam based outer update direction as computed in Algorithm 2.

In Figure 4, we plot the training and testing performance with the number of iterations for SUSTAIN and other algorithms for 5-way 5-shot learning task on `FC100` dataset. Note that SUSTAIN outperforms rest of the algorithms on the training task and performs on par with other algorithms with respect to the testing performance. Moreover, note that initially ANIL performs better but since the number of inner steps are relatively large (20 in this case), ANIL's performance degrades after a certain number of iterations. Similar behavior was noted for ANIL in the results of [19].

The above set of experiments showed that the Adam [21] optimizer can be incorporated with SUSTAIN and other algorithms to achieve improved performance compared to vanilla SG based algorithms. We also showed that the gradient estimator for SUSTAIN can be modified to require only single (batch) gradient evaluation per iteration (cf. (20)) without comprising performance of the algorithm. In this section, we use an additional set of results to demonstrate that under most settings SUSTAIN outperforms other state-of-the-art algorithms.

**Hyperparameter optimization.** For data hyperparameter optimization problem, we consider the hyper-cleaning task as discussed earlier in Section 4 and benchmark the performance of SUSTAIN against stocBiO [19] and HOAG [29]. Importantly, in this section we demonstrate that SUSTAIN performs well under (relatively) high level of data corruption.

Here we consider an additional set of results for the hyper-cleaning task on Fashion-MNIST dataset [41]. All the parameter settings are the same as in Section 4, except that we use a higher level 40% corruption rate. Note that HOAG is a deterministic algorithm and requires full gradient computation at each iteration. In contrast, stocBiO is a stochastic algorithm but it relies on large batch gradient computations. We conduct experiments for two settings where stocBiO uses a batch size of 5000 and 1000 (for both inner and outer updates). Our algorithm SUSTAIN is purely a stochastic algorithm and does not rely on large batch gradient computations. Specifically, SUSTAIN computes two gradients (on a single sample) in each iteration for both inner and outer updates (cf. (13) and (14)). Since at

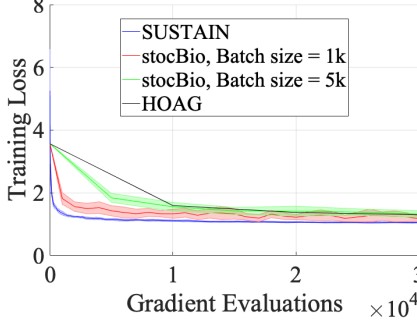
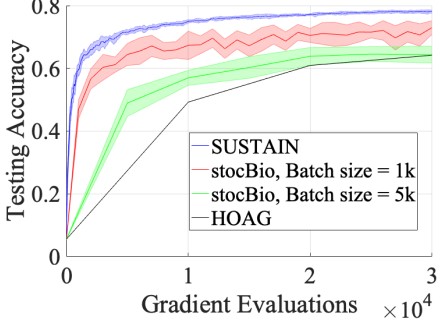

Figure 5: Data hyperparameter optimization: Training loss and testing accuracy against the number of gradients evaluated with corruption rate $p = 0.4$.

each outer iteration, the sample sizes (and gradient computations) accessed by each algorithms are very different, so it is no longer fair to compare the per-iteration performance for different algorithms (this is different compared with the meta learning example in the previous section). Therefore, in this section we compare the training and testing performance of the competing algorithms using the number of total outer gradient computations (which is same as the inner gradient computations) across iterations. Note that for HOAG and stocBiO, the number of samples accessed is same as the number of gradient evaluations, whereas for SUSTAIN we compute two gradients for each sample accessed (cf. (14))[3]. The experiments in Figure 5 establish that SUSTAIN outperforms HOAG and stocBiO, in terms of the total number of gradient evaluations as well as the number of samples, even under high corruption rate.

---

[3]Note that this requirement can be easily relaxed without compromising performance via using the gradient construction (20).

## Proofs of Theoretical Results

Now we present the proofs of the theoretical results.

## B   Useful lemmas

**Lemma B.1.** *Consider a collection of functions $\Phi_i : \mathbb{R}^n \to \mathcal{Z}$ with $i = \{1, 2, \ldots, k\}$ and $\mathcal{Z} \subseteq \mathbb{R}^{n \times n}$, which satisfy the following assumptions:*

*(i) There exist $L_i > 0$, $i \in [k]$, such that*

$$\|\Phi_i(x) - \Phi_i(y)\| \le L_i \|x - y\|, \ \forall \, i \in [k], \ x, y \in \mathbb{R}^n.$$

*(ii) For each $i \in [N]$ and $k \in \mathbb{N}$ we have $\|\Phi_i(x)\| \le M_i$ for all $x \in \mathbb{R}^n$.*

*Then the following holds for all $x, y \in \mathbb{R}^n$:*

$$\left\| \prod_{i=1}^{k} \Phi_i(x) - \prod_{i=1}^{k} \Phi_i(y) \right\|^2 \le k \sum_{i=1}^{k} \left( \prod_{j=1, j \neq i}^{k} M_j \right)^2 L_i^2 \|x - y\|^2. \tag{21}$$

*Moreover, if $k$ is generated uniformly at random from $\{0, 1, \ldots, K-1\}$, then the following holds for all $x, y \in \mathbb{R}^n$:*

$$\mathbb{E}_k \left\| \prod_{i=1}^{k} \Phi_i(x) - \prod_{i=1}^{k} \Phi_i(y) \right\|^2 \le K \sum_{i=1}^{K} \mathbb{E}_k \left[ \left( \prod_{j=1, j \neq i}^{k} M_j \right)^2 \right] L_i^2 \|x - y\|^2. \tag{22}$$

*Here we use the convention that $\prod_{i=1}^{k} \Phi_i(x) = I$ if $k = 0$.*

*Proof.* We first prove (21). To do so we will first show that the following holds for all $x, y \in \mathbb{R}^n$ and $k \in \mathbb{N}$:

$$\left\| \prod_{i=1}^{k} \Phi_i(x) - \prod_{i=1}^{k} \Phi_i(y) \right\| \le \sum_{i=1}^{k} \left( \prod_{j=1, j \neq i}^{k} M_j \right) L_i \|x - y\|, \tag{23}$$

Then by combining the above result with the identity that

$$\|z_1 + z_2 + \ldots + z_k\|^2 \le k \|z_1\|^2 + k \|z_2\|^2 + \ldots + k \|z_k\|^2, \text{ for all } z, \ k \in \mathbb{N}, \tag{24}$$

we can conclude the first statement.

To show (23), we use an induction argument. The base case for $k = 1$ holds because of the Lipschitz assumption $(i)$ given in the statement of the lemma. Then assuming claim (23) holds for arbitrary $k$, we have for $k + 1$

$$\left\| \prod_{i=1}^{k+1} \Phi_i(x) - \prod_{i=1}^{k+1} \Phi_i(y) \right\| = \left\| \prod_{i=1}^{k+1} \Phi_i(x) - \prod_{i=1}^{k} \Phi_i(x)\Phi_{k+1}(y) + \prod_{i=1}^{k} \Phi_i(x)\Phi_{k+1}(y) - \prod_{i=1}^{k+1} \Phi_i(y) \right\|$$

$$\overset{(a)}{\le} \left\| \prod_{i=1}^{k} \Phi_i(x) \right\| \left\| \Phi_{k+1}(x) - \Phi_{k+1}(y) \right\| + \left\| \Phi_{k+1}(y) \right\| \left\| \prod_{i=1}^{k} \Phi_i(x) - \prod_{i=1}^{k} \Phi_i(y) \right\|$$

$$\overset{(b)}{\le} \left( \prod_{j=1}^{k} M_j \right) L_{k+1} \|x - y\| + \sum_{i=1}^{k} \left( \prod_{j=1, j \neq i}^{k+1} M_j \right) L_i \|x - y\|$$

$$\overset{(c)}{\le} \sum_{i=1}^{k+1} \left( \prod_{j=1, j \neq i}^{k+1} M_j \right) L_i \|x - y\|.$$

where $(a)$ follows from the application of the triangle inequality and the Cauchy-Schwartz inequality; the first expression in $(b)$ results from the application of Cauchy-Schwartz inequality and Assumption (i) and (ii) of the statement of the lemma; the second expression in $(b)$ follows from the assumption

that claim (23) holds for $k$; $(c)$ follows from combining the two expressions. We conclude that (23) holds for all $k \in \mathbb{N}$.

Now consider the case when $k$ is chosen uniformly at random from $k \in \{0, 1, \ldots, K-1\}$. First, note from the definition that for $k = 0$ we have $\prod_{i=1}^{k} \Phi_i(x) = I$. This implies that (21) is also satisfied if we have $k = 0$. We then have

$$
\mathbb{E}_k \left\| \prod_{i=1}^{k} \Phi_i(x) - \prod_{i=1}^{k} \Phi_i(y) \right\|^2 \overset{(a)}{\le} \mathbb{E}_k \left[ k \sum_{i=1}^{k} \Big( \prod_{j=1, j \neq i}^{k} M_j \Big)^2 \|\Phi_i(x) - \Phi_i(y)\|^2 \right]
$$

$$
\overset{(b)}{\le} K \sum_{i=1}^{K} \mathbb{E}_k \left[ \Big( \prod_{j=1, j \neq i}^{k} M_j \Big)^2 \right] \|\Phi_i(x) - \Phi_i(y)\|^2
$$

$$
\overset{(c)}{\le} K \sum_{i=1}^{K} \mathbb{E}_k \left[ \Big( \prod_{j=1, j \neq i}^{k} M_j \Big)^2 \right] L_i^2 \|x - y\|^2.
$$

where $(a)$ uses the fact that (21) holds for all $k \in \{0, 1, \ldots, K-1\}$ almost surely; $(b)$ follows from the fact that $k \le K$ almost surely; $(c)$ results from Assumption $(i)$ of the lemma. $\square$

## C Proofs of preliminary lemmas

### C.1 Estimation of the stochastic gradient

We construct the stochastic gradient $\bar{\nabla} f(x, y; \bar{\xi})$ as [14, 18]:

1. For $K \in \mathbb{N}$, choose $k \in \{0, 1, \ldots, K-1\}$ uniformly at random.
2. Compute unbiased Hessian approximations $\nabla_{xy}^2 g(x, y; \zeta^{(0)})$ and $\nabla_{yy}^2 g(x, y; \zeta^{(i)})$ for $i \in \{1, \ldots, k\}$, where $\{\zeta^{(i)}\}_{i=0}^{k}$ are chosen independently.
3. Compute unbiased gradient approximations $\nabla_x f(x, y; \xi)$ and $\nabla_y f(x, y; \xi)$ where $\xi$ is chosen independently of $\{\zeta^{(i)}\}_{i=0}^{k}$.
4. Construct the stochastic gradient estimate $\bar{\nabla} f(x, y; \bar{\xi})$ with $\bar{\xi}$ denoted as $\bar{\xi} = \{\xi, \{\zeta^{(i)}\}_{i=0}^{k}\}$:
$$
\bar{\nabla} f(x, y; \bar{\xi})
$$
$$
= \nabla_x f(x, y; \xi) - \nabla_{xy}^2 g(x, y; \zeta^{(0)}) \left[ \frac{K}{L_g} \prod_{i=1}^{k} \left( I - \frac{1}{L_g} \nabla_{yy}^2 g(x, y; \zeta^{(i)}) \right) \right] \nabla_y f(x, y; \xi),
$$
(25)

   with $\prod_{i=1}^{k} \left( I - \frac{1}{L_g} \nabla_{yy}^2 g(x, y; \zeta^{(i)}) \right) = I$ if $k = 0$.

Next, we state the result showing that the bias of the stochastic gradient estimate of the upper level objective defined in (7) decays linearly with the number of samples $K$ chosen to approximate the Hessian inverse.

**Lemma C.1.** *[18, Lemma 11] Under Assumptions 1, 2 and 3 the stochastic gradient estimate of the upper level objective defined in (25), satisfies*

$$
\|B(x, y)\| = \|\bar{\nabla} f(x, y) - \mathbb{E}[\bar{\nabla} f(x, y; \bar{\xi})]\| \le \frac{C_{g_{xy}} C_{f_y}}{\mu_g} \left( 1 - \frac{\mu_g}{L_g} \right)^K,
$$

*where $B(x, y)$ is the bias of the stochastic gradient estimate and $K$ is the number of samples chosen to approximate the Hessian inverse in (25). Moreover, we have*

$$
\mathbb{E}_{\bar{\xi}} \left[ \left\| \bar{\nabla} f(x, y) - \mathbb{E}_{\bar{\xi}} [\bar{\nabla} f(x, y; \bar{\xi})] \right\|^2 \right] \le \sigma_{f_x}^2 + \frac{3}{\mu_g^2} \left[ (\sigma_{f_y}^2 + C_{f_y}^2)(\sigma_{g_{xy}}^2 + 2C_{g_{xy}}^2) + \sigma_{f_y}^2 C_{g_{xy}}^2 \right].
$$

Lemma C.1 implies that the bias $B(x, y)$ can be made to satisfy $\|B(x, y)\| \le \epsilon$ with only

$$
K = (L_g / \mu_g) \log(C_{g_{xy}} C_{f_y} / \mu_g \epsilon)
$$

stochastic Hessian samples of $\nabla_{yy}^2 g(x, y)$.

## C.2 Lipschitz continuity of gradient estimate

**Lemma C.2** (Lipschitzness of Stochastic Gradient Estimate). *If the stochastic functions $f(x, y; \xi)$ and $g(x, y; \zeta)$ satisfy Assumptions 1, 2 and 3, then we have*

*(i) For a fixed $y \in \mathbb{R}^{d_{\mathsf{up}}}$*

$$\mathbb{E}_{\bar{\xi}}\|\bar{\nabla} f(x_1, y; \bar{\xi}) - \bar{\nabla} f(x_2, y; \bar{\xi})\|^2 \le L_K^2 \|x_1 - x_2\|^2, \ \forall \ x_1, x_2 \in \mathbb{R}^{d_{\mathsf{up}}}.$$

*(ii) For a fixed $x \in \mathbb{R}^{d_{\mathsf{up}}}$*

$$\mathbb{E}_{\bar{\xi}}\|\bar{\nabla} f(x, y_1; \bar{\xi}) - \bar{\nabla} f(x, y_2; \bar{\xi})\|^2 \le L_K^2 \|y_1 - y_2\|^2, \ \forall \ y_1, y_2 \in \mathbb{R}^{d_{\mathsf{up}}}.$$

*In the above expressions, $L_K > 0$ is defined as:*

$$L_K^2 = 2L_{f_x}^2 + 6C_{g_{xy}}^2 L_{f_y}^2 \left( \frac{K}{2\mu_g L_g - \mu_g^2} \right) + 6C_{f_y}^2 L_{g_{xy}}^2 \left( \frac{K}{2\mu_g L_g - \mu_g^2} \right)$$
$$+ 6C_{g_{xy}}^2 C_{f_y}^2 \frac{K^3 L_g^2}{(L_g - \mu_g)^2 (2\mu_g L_g - \mu_g^2)},$$

*and where $K$ is the number of samples required to construct the stochastic approximation of $\bar{\nabla} f$ (see (25) above).*

*Proof.* We prove only statement $(i)$ of the lemma, the proof of $(ii)$ follows from a similar argument. From the definition of $\bar{\nabla} f(x_1, y; \bar{\xi})$ we have for $x_1, x_2 \in \mathbb{R}^{d_{\mathsf{up}}}$ and $y \in \mathbb{R}^{d_{\mathsf{up}}}$

$$\|\bar{\nabla} f(x_1, y; \bar{\xi}) - \bar{\nabla} f(x_2, y; \bar{\xi})\|^2$$
$$\overset{(a)}{\le} 2\left\|\nabla_x f(x_1, y; \xi) - \nabla_x f(x_2, y; \xi)\right\|^2$$
$$+ 2\left\|\nabla_{xy}^2 g(x_1, y; \zeta^{(0)}) \left[ \frac{K}{L_g} \prod_{i=1}^{k} \left( I - \frac{1}{L_g} \nabla_{yy}^2 g(x_1, y; \zeta^{(i)}) \right) \right] \nabla_y f(x_1, y; \xi) \right.$$
$$\left. - \nabla_{xy}^2 g(x_2, y; \zeta^{(0)}) \left[ \frac{K}{L_g} \prod_{i=1}^{k} \left( I - \frac{1}{L_g} \nabla_{yy}^2 g(x_2, y; \zeta^{(i)}) \right) \right] \nabla_y f(x_2, y; \xi) \right\|^2$$
$$\overset{(b)}{\le} 2L_{f_x}^2 \left\|x_1 - x_2\right\|^2$$
$$+ 2\left\|\nabla_{xy}^2 g(x_1, y; \zeta^{(0)}) \left[ \frac{K}{L_g} \prod_{i=1}^{k} \left( I - \frac{1}{L_g} \nabla_{yy}^2 g(x_1, y; \zeta^{(i)}) \right) \right] \nabla_y f(x_1, y; \xi) \right.$$
$$\left. - \nabla_{xy}^2 g(x_2, y; \zeta^{(0)}) \left[ \frac{K}{L_g} \prod_{i=1}^{k} \left( I - \frac{1}{L_g} \nabla_{yy}^2 g(x_2, y; \zeta^{(i)}) \right) \right] \nabla_y f(x_2, y; \xi) \right\|^2, \quad (26)$$

where inequality $(a)$ follows from the definition of $\bar{\nabla} f(x_1, y; \bar{\xi})$ and (24); inequality $(b)$ follows from the Lipschitz-ness Assumption 1–(ii) made for stochastic upper level objective. The variable $k \in \{0, \dots, K-1\}$ above is a random variable define in Section C.1 above. Let us consider the second term of (26) above, we have

$$\left\|\nabla_{xy}^2 g(x_1, y; \zeta^{(0)}) \left[ \frac{K}{L_g} \prod_{i=1}^{k} \left( I - \frac{1}{L_g} \nabla_{yy}^2 g(x_1, y; \zeta^{(i)}) \right) \right] \nabla_y f(x_1, y; \xi) \right.$$
$$\left. - \nabla_{xy}^2 g(x_2, y; \zeta^{(0)}) \left[ \frac{K}{L_g} \prod_{i=1}^{k} \left( I - \frac{1}{L_g} \nabla_{yy}^2 g(x_2, y; \zeta^{(i)}) \right) \right] \nabla_y f(x_2, y; \xi) \right\|^2$$
$$\overset{(a)}{\le} 3C_{g_{xy}}^2 \frac{K^2}{L_g^2} \left( 1 - \frac{\mu_g}{L_g} \right)^{2k} \|\nabla_y f(x_1, y; \xi) - \nabla_y f(x_2, y; \xi)\|^2$$

$$+ 3C_{f_y}^2 \frac{K^2}{L_g^2} \left(1 - \frac{\mu_g}{L_g}\right)^{2k} \|\nabla_{xy}^2 g(x_1, y; \zeta^{(0)}) - \nabla_{xy}^2 g(x_2, y; \zeta^{(0)})\|^2$$

$$+ 3C_{g_{xy}}^2 C_{f_y}^2 \left\| \frac{K}{L_g} \prod_{i=1}^{k} \left(I - \frac{1}{L_g} \nabla_{yy}^2 g(x_1, y; \zeta^{(i)})\right) - \frac{K}{L_g} \prod_{i=1}^{k} \left(I - \frac{1}{L_g} \nabla_{yy}^2 g(x_2, y; \zeta^{(i)})\right) \right\|^2$$

$$\overset{(b)}{\leq} 3C_{g_{xy}}^2 \frac{K^2}{L_g^2} \left(1 - \frac{\mu_g}{L_g}\right)^{2k} L_{f_y}^2 \|x_1 - x_2\|^2 + 3C_{f_y}^2 \frac{K^2}{L_g^2} \left(1 - \frac{\mu_g}{L_g}\right)^{2k} L_{g_{xy}}^2 \|x_1 - x_2\|^2$$

$$+ 3C_{g_{xy}}^2 C_{f_y}^2 \frac{K^2}{L_g^2} \left\| \prod_{i=1}^{k} \left(I - \frac{1}{L_g} \nabla_{yy}^2 g(x_1, y; \zeta^{(i)})\right) - \prod_{i=1}^{k} \left(I - \frac{1}{L_g} \nabla_{yy}^2 g(x_2, y; \zeta^{(i)})\right) \right\|^2,$$

where inequality $(a)$ follows from (21) in Lemma B.1, Assumption 1–(iii) and Assumption 2–(ii)(iii)(vi); inequality $(b)$ follows from the Lipschitz continuity Assumption 1–(ii) and Assumption 2–(v) made for the stochastic upper and lower level objectives. On both sides taking expectation w.r.t $k$, we get:

$$\mathbb{E}_k \left\| \nabla_{xy}^2 g(x_1, y; \zeta^{(0)}) \left[ \frac{K}{L_g} \prod_{i=1}^{k} \left(I - \frac{1}{L_g} \nabla_{yy}^2 g(x_1, y; \zeta^{(i)})\right) \right] \nabla_y f(x_1, y; \xi) \right.$$

$$\left. - \nabla_{xy}^2 g(x_2, y; \zeta^{(0)}) \left[ \frac{K}{L_g} \prod_{i=1}^{k} \left(I - \frac{1}{L_g} \nabla_{yy}^2 g(x_2, y; \zeta^{(i)})\right) \right] \nabla_y f(x_2, y; \xi) \right\|^2$$

$$\leq 3C_{g_{xy}}^2 \frac{K^2}{L_g^2} \mathbb{E}_k \left[ \left(1 - \frac{\mu_g}{L_g}\right)^{2k} \right] L_{f_y}^2 \|x_1 - x_2\|^2 + 3C_{f_y}^2 \frac{K^2}{L_g^2} \mathbb{E}_k \left[ \left(1 - \frac{\mu_g}{L_g}\right)^{2k} \right] L_{g_{xy}}^2 \|x_1 - x_2\|^2$$

$$+ 3C_{g_{xy}}^2 C_{f_y}^2 \frac{K^2}{L_g^2} \mathbb{E}_k \left\| \prod_{i=1}^{k} \left(I - \frac{1}{L_g} \nabla_{yy}^2 g(x_1, y; \zeta^{(i)})\right) - \prod_{i=1}^{k} \left(I - \frac{1}{L_g} \nabla_{yy}^2 g(x_2, y; \zeta^{(i)})\right) \right\|^2$$

$$\overset{(a)}{\leq} 3C_{g_{xy}}^2 L_{f_y}^2 \left(\frac{K}{2\mu_g L_g - \mu_g^2}\right) \|x_1 - x_2\|^2 + 3C_{f_y}^2 L_{g_{xy}}^2 \left(\frac{K}{2\mu_g L_g - \mu_g^2}\right) \|x_1 - x_2\|^2$$

$$+ 3C_{g_{xy}}^2 C_{f_y}^2 \frac{K^2}{L_g^2} \mathbb{E}_k \left\| \prod_{i=1}^{k} \left(I - \frac{1}{L_g} \nabla_{yy}^2 g(x_1, y; \zeta^{(i)})\right) - \prod_{i=1}^{k} \left(I - \frac{1}{L_g} \nabla_{yy}^2 g(x_2, y; \zeta^{(i)})\right) \right\|^2,$$
$$\tag{27}$$

where $(a)$ follows from the fact that we have:

$$\mathbb{E}_k \left[ \left(1 - \frac{\mu_g}{L_g}\right)^{2k} \right] = \frac{1}{K} \sum_{k=0}^{K-1} \left(1 - \frac{\mu_g}{L_g}\right)^{2k} \leq \frac{1}{K} \left(\frac{L_g^2}{2\mu_g L_g - \mu_g^2}\right),$$

where the first equality above follows from the fact that $k \in \{0, 1, \ldots, K-1\}$ is chosen uniformly at random and the second equality results from the sum of a geometric progression.

Finally, considering the last term of (27), we have

$$\mathbb{E}_k \left\| \prod_{i=1}^{k} \left(I - \frac{1}{L_g} \nabla_{yy}^2 g(x_1, y; \zeta^{(i)})\right) - \prod_{i=1}^{k} \left(I - \frac{1}{L_g} \nabla_{yy}^2 g(x_2, y; \zeta^{(i)})\right) \right\|^2$$

$$\overset{(a)}{\leq} K \sum_{i=1}^{K} \mathbb{E}_k \left[ \left(1 - \frac{\mu_g}{L_g}\right)^{2(k-1)} \right] \frac{1}{L_g^2} \|\nabla_{yy}^2 g(x_1, y; \zeta^{(i)}) - \nabla_{yy}^2 g(x_2, y; \zeta^{(i)})\|^2$$

$$\overset{(b)}{\leq} \left(\frac{L_g^2}{(L_g - \mu_g)^2}\right) \left(\frac{1}{2\mu_g L_g - \mu_g^2}\right) \sum_{i=1}^{K} \|\nabla_{yy}^2 g(x_1, y; \zeta^{(i)}) - \nabla_{yy}^2 g(x_2, y; \zeta^{(i)})\|^2$$

$$\overset{(c)}{\leq} \frac{K L_g^2 L_{g_{yy}}^2}{(L_g - \mu_g)^2 (2\mu_g L_g - \mu_g^2)} \|x_1 - x_2\|^2,$$
$$\tag{28}$$

where $(a)$ follows from the application of (22) in Lemma B.1 along with Assumption 2–(ii)(iii); inequality $(b)$ utilizes

$$\mathbb{E}_k\left[\left(1 - \frac{\mu_g}{L_g}\right)^{2(k-1)}\right] = \frac{1}{K}\sum_{k=0}^{K-1}\left(1 - \frac{\mu_g}{L_g}\right)^{2(k-1)} \leq \frac{1}{K}\left(\frac{L_g^2}{(L_g - \mu_g)^2}\right)\left(\frac{L_g^2}{2\mu_g L_g - \mu_g^2}\right),$$

where the first equality above again utilizes the fact that $k \in \{0, 1, \ldots, K-1\}$ is chosen uniformly at random and the second equality results from the sum of a geometric progression; inequality $(c)$ utilizes Assumption 2–(v) made for stochastic lower level objective.

Finally, taking expectation in (26) and substituting the expressions obtained in (27) and (28) in (26), we obtain

$$\mathbb{E}\|\bar{\nabla}f(x_1, y; \bar{\xi}) - \bar{\nabla}f(x_2, y; \bar{\xi})\|^2 \leq L_K^2\|x_1 - x_2\|^2,$$

where $L_K^2$ defined as:

$$L_K^2 := 2L_{f_x}^2 + 6C_{g_{xy}}^2 L_{f_y}^2\left(\frac{K}{2\mu_g L_g - \mu_g^2}\right) + 6C_{f_y}^2 L_{g_{xy}}^2\left(\frac{K}{2\mu_g L_g - \mu_g^2}\right)$$
$$+ 6C_{g_{xy}}^2 C_{f_y}^2 \frac{K^3 L_{g_{yy}}^2}{(L_g - \mu_g)^2(2\mu_g L_g - \mu_g^2)}.$$

Statement $(i)$ of the Lemma is proved.

The proof of the statement $(ii)$ follows the same procedure, so it is omitted. $\qquad\square$

# D  Proof of Theorem 3.2: smooth (possibly non-convex) outer objective

First, we consider the descent achieved by the outer objective in consecutive iterates generated by the Algorithm 1 when the outer problem is smooth and is possibly non-convex. We define the following constants for the stepsize parameters:

$$w = \max\left\{2,\ 27L_f^3,\ 8L_{\mu_g}^3 c_\beta^3,\ (\mu_g + L_g)^3 c_\beta^3,\ c_{\eta_f}^{3/2},\ c_{\eta_g}^{3/2}\right\}, \quad c_\beta = \frac{6\sqrt{2}L_y L}{L_{\mu_g}},$$

$$c_{\eta_f} = \frac{1}{3L_f} + \max\left\{36L_K^2, \frac{4L_K^2 L_{\mu_g}(\mu_g + L_g)c_\beta^2}{L^2}\right\}, \tag{29}$$

$$c_{\eta_g} = \frac{1}{3L_f} + 8L_g^2 c_\beta^2 + \left[\frac{8L^2}{L_{\mu_g}^2} + \frac{2L^2}{L_{\mu_g}(\mu_g + L_g)}\right]\max\left\{36L_g^2, \frac{4L_g^2 L_{\mu_g}(\mu_g + L_g)c_\beta^2}{L^2}\right\},$$

where we have defined $L_{\mu_g} = \frac{\mu_g L_g}{\mu_g + L_g}$.

## D.1  Descent in the function value

**Lemma D.1.** *For non-convex and smooth $\ell(\cdot)$, with $e_t^f$ defined as: $e_t^f := h_t^f - \bar{\nabla}f(x_t, y_t) - B_t$, the consecutive iterates of Algorithm 1 satisfy:*

$$\mathbb{E}[\ell(x_{t+1})] \leq \mathbb{E}\Big[\ell(x_t) - \frac{\alpha_t}{2}\|\nabla\ell(x_t)\|^2 - \frac{\alpha_t}{2}(1 - \alpha_t L_f)\|h_t^f\|^2 + \alpha_t\|e_t^f\|^2$$
$$+ 2\alpha_t L^2\|y_t - y^*(x_t)\|^2 + 2\alpha_t\|B_t\|^2\Big].$$

*for all $t \in \{0, 1, \ldots, T-1\}$, where the expectation is w.r.t. the stochasticity of the algorithm.*

*Proof.* Using the Lipschitz smoothness of the objective function from Lemma 2.2 we have:

$$\ell(x_{t+1}) \leq \ell(x_t) + \langle\nabla\ell(x_t), x_{t+1} - x_t\rangle + \frac{L_f}{2}\|x_{t+1} - x_t\|^2$$

$$\overset{(a)}{=} \ell(x_t) - \alpha_t\langle\nabla\ell(x_t), h_t^f\rangle + \frac{\alpha_t^2 L_f}{2}\|h_t^f\|^2$$

$$\overset{(b)}{=} \ell(x_t) - \frac{\alpha_t}{2}\|\nabla\ell(x_t)\|^2 - \frac{\alpha_t}{2}(1 - \alpha_t L_f)\|h_t^f\|^2 + \frac{\alpha_t}{2}\|h_t^f - \nabla\ell(x_t)\|^2. \tag{30}$$

where $(a)$ results from Step 7 of Algorithm 1 and $(b)$ uses $\langle a, b \rangle = \frac{1}{2}\|a\|^2 + \frac{1}{2}\|b\|^2 - \frac{1}{2}\|a - b\|^2$. Next, we bound the term $\|h_t^f - \nabla\ell(x_t)\|^2$ as follows

$$
\begin{aligned}
\|h_t^f - \nabla\ell(x_t)\|^2 &= \|h_t^f - \bar{\nabla}f(x_t, y_t) - B_t + \bar{\nabla}f(x_t, y_t) + B_t - \nabla\ell(x_t)\|^2 \\
&\overset{(c)}{\leq} 2\|h_t^f - \bar{\nabla}f(x_t, y_t) - B_t\|^2 + 4\|\bar{\nabla}f(x_t, y_t) - \nabla\ell(x_t)\|^2 + 4\|B_t\|^2 \\
&\overset{(d)}{\leq} 2\|e_t^f\|^2 + 4L^2\|y_t - y^*(x_t)\|^2 + 4\|B_t\|^2,
\end{aligned}
$$

where inequality $(c)$ uses (24) and $(d)$ results from the definition of $e_t^f := h_t^f - \bar{\nabla}f(x_t, y_t) - B_t$ and (11) in Lemma 2.2. Substituting the above in (30) and taking expectation w.r.t. the stochasticity of the algorithm we get the statement of the lemma. $\qquad\square$

### D.2  Descent in the iterates of the lower level problem

**Lemma D.2.** *Define $e_t^g := h_t^g - \nabla_y g(x_t, y_t)$. then the iterates of the inner problem generated according to Algorithm 1, satisfy*

$$
\begin{aligned}
&\mathbb{E}\|y_{t+1} - y^*(x_{t+1})\|^2 \\
&\leq (1+\gamma_t)(1+\delta_t)\left(1 - 2\beta_t \frac{\mu_g L_g}{\mu_g + L_g}\right)\mathbb{E}\|y_t - y^*(x_t)\|^2 + \left(1 + \frac{1}{\gamma_t}\right)L_y^2\alpha_t^2\mathbb{E}\|h_t^f\|^2 \\
&\quad - (1+\gamma_t)(1+\delta_t)\left(\frac{2\beta_t}{\mu_g + L_g} - \beta_t^2\right)\mathbb{E}\|\nabla_y g(x_t, y_t)\| + (1+\gamma_t)\left(1 + \frac{1}{\delta_t}\right)\beta_t^2\mathbb{E}\|e_t^g\|^2.
\end{aligned}
$$

*for all $t \in \{0, \ldots, T-1\}$ with some $\gamma_t, \delta_t > 0$., where the expectation is w.r.t. the stochasticity of the algorithm.*

*Proof.* Consider the term $\mathbb{E}\|y_{t+1} - y^*(x_{t+1})\|^2$, we have

$$
\begin{aligned}
\mathbb{E}\|y_{t+1} - y^*(x_{t+1})\|^2 &\overset{(a)}{\leq} (1+\gamma_t)\mathbb{E}\|y_{t+1} - y^*(x_t)\|^2 + \left(1 + \frac{1}{\gamma_t}\right)\mathbb{E}\|y^*(x_t) - y^*(x_{t+1})\|^2 \\
&\overset{(b)}{=} (1+\gamma_t)\mathbb{E}\|y_t - \beta_t h_t^g - y^*(x_t)\|^2 + \left(1 + \frac{1}{\gamma_t}\right)L_y^2\mathbb{E}\|x_{t+1} - x_t\|^2 \\
&\overset{(c)}{\leq} (1+\gamma_t)(1+\delta_t)\mathbb{E}\|y_t - \beta_t\nabla_y g(x_t, y_t) - y^*(x_t)\|^2 \\
&\quad + (1+\gamma_t)\left(1 + \frac{1}{\delta_t}\right)\beta_t^2\|h_t^g - \nabla_y g(x_t, y_t)\|^2 + \left(1 + \frac{1}{\gamma_t}\right)L_y^2\alpha_t^2\mathbb{E}\|h_t^f\|^2.
\end{aligned}
$$
$$(31)$$

where $(a)$ results from the Young's inequality; $(b)$ uses Step 5 of Algorithm 1 and Lipschitzness of $y^*(\cdot)$ in Lemma 2.2 and $(c)$ again utilizes Young's inequality and Step 7 of Algorithm 1. Next, we consider the first term of the above equation we have

$$
\begin{aligned}
&\|y_t - \beta_t\nabla_y g(x_t, y_t) - y^*(x_t)\|^2 \\
&= \|y_t - y^*(x_t)\|^2 + \beta_t^2\|\nabla_y g(x_t, y_t)\|^2 - 2\beta_t\langle\nabla_y g(x_t, y_t), y_t - y^*(x_t)\rangle \\
&\overset{(d)}{\leq} \left(1 - 2\beta_t\frac{\mu_g L_g}{\mu_g + L_g}\right)\|y_t - y^*(x_t)\|^2 - \left(\frac{2\beta_t}{\mu_g + L_g} - \beta_t^2\right)\|\nabla_y g(x_t, y_t)\|^2,
\end{aligned}
$$

where inequality $(d)$ above results from the strong convexity of $g$, which implies

$$
\langle\nabla_y g(x_t, y_t), y_t - y^*(x_t)\rangle \geq \frac{\mu_g L_g}{\mu_g + L_g}\|y_t - y^*(x_t)\|^2 + \frac{1}{\mu_g + L_g}\|\nabla_y g(x_t, y_t)\|^2.
$$

Substituting in (31) and using the definition $e_t^g := h_t^g - \nabla_y g(x_t, y_t)$ we get the statement of the lemma. $\qquad\square$

## D.3 Descent in the gradient estimation error of the outer function

Before presenting the descent in the gradient estimation error of the outer function we define $\mathcal{F}_t = \sigma\{y_0, x_0, \ldots, y_t, x_t\}$ as the sigma algebra generated by the sequence of iterates up to the $t$th iteration of SUSTAIN.

**Lemma D.3.** *Define $e_t^f := h_t^f - \bar{\nabla} f(x_t, y_t) - B_t$. Then the consecutive iterates of Algorithm 1 satisfy:*

$$\mathbb{E}\|e_{t+1}^f\|^2 \leq (1 - \eta_{t+1}^f)^2 \mathbb{E}\|e_t^f\|^2 + 2(\eta_{t+1}^f)^2 \sigma_f^2 + 4(1 - \eta_{t+1}^f)^2 L_K^2 \alpha_t^2 \mathbb{E}\|h_t^f\|^2$$
$$+ 8(1 - \eta_{t+1}^f)^2 L_K^2 \beta_t^2 \mathbb{E}\|e_t^g\|^2 + 8(1 - \eta_{t+1}^f)^2 L_K^2 \beta_t^2 \mathbb{E}\|\nabla_y g(x_t, y_t)\|^2,$$

*for all $t \in \{0, \ldots, T-1\}$, with $L_K$ defined in the statement of Lemma C.2. Here the expectation is taken w.r.t the stochasticity of the algorithm.*

*Proof.* From the definition of $e_t^f$ we have

$$\mathbb{E}\|e_{t+1}^f\|^2 \tag{32}$$
$$= \mathbb{E}\|h_{t+1}^f - \bar{\nabla} f(x_{t+1}, y_{t+1}) - B_{t+1}\|^2$$
$$\overset{(a)}{=} \mathbb{E}\|\eta_{t+1}^f \bar{\nabla} f(x_{t+1}, y_{t+1}; \xi_{t+1}) + (1 - \eta_{t+1})\big(h_t^f + \bar{\nabla} f(x_{t+1}, y_{t+1}; \xi_{t+1}) - \bar{\nabla} f(x_t, y_t; \xi_{t+1})\big)$$
$$- \bar{\nabla} f(x_{t+1}, y_{t+1}) - B_{t+1}\|^2$$
$$\overset{(b)}{=} \mathbb{E}\|(1 - \eta_{t+1}^f) e_t^f + \eta_{t+1}^f(\bar{\nabla} f(x_{t+1}, y_{t+1}; \xi_{t+1}) - \bar{\nabla} f(x_{t+1}, y_{t+1}) - B_{t+1})$$
$$+ (1 - \eta_{t+1}^f)\big((\bar{\nabla} f(x_{t+1}, y_{t+1}; \xi_{t+1}) - \bar{\nabla} f(x_{t+1}, y_{t+1}) - B_{t+1})$$
$$- (\bar{\nabla} f(x_t, y_t; \xi_{t+1}) - \bar{\nabla} f(x_t, y_t) - B_t)\big)\|^2$$
$$\overset{(c)}{=} (1 - \eta_{t+1}^f)^2 \mathbb{E}\|e_t^f\|^2 + \mathbb{E}\|\eta_{t+1}^f(\bar{\nabla} f(x_{t+1}, y_{t+1}; \xi_{t+1}) - \bar{\nabla} f(x_{t+1}, y_{t+1}) - B_{t+1})$$
$$+ (1 - \eta_{t+1}^f)\big((\bar{\nabla} f(x_{t+1}, y_{t+1}; \xi_{t+1}) - \bar{\nabla} f(x_{t+1}, y_{t+1}) - B_{t+1})$$
$$- (\bar{\nabla} f(x_t, y_t; \xi_{t+1}) - \bar{\nabla} f(x_t, y_t) - B_t)\big)\|^2$$
$$\overset{(d)}{\leq} (1 - \eta_{t+1}^f)^2 \mathbb{E}\|e_t^f\|^2 + 2(\eta_{t+1}^f)^2 \mathbb{E}\|\bar{\nabla} f(x_{t+1}, y_{t+1}; \xi_{t+1}) - \bar{\nabla} f(x_{t+1}, y_{t+1}) - B_{t+1}\|^2$$
$$+ 2(1 - \eta_t^f)^2 \mathbb{E}\|(\bar{\nabla} f(x_{t+1}, y_{t+1}; \xi_{t+1}) - \bar{\nabla} f(x_{t+1}, y_{t+1}) - B_{t+1})$$
$$- (\bar{\nabla} f(x_t, y_t; \xi_{t+1}) - \bar{\nabla} f(x_t, y_t) - B_t)\|^2$$
$$\overset{(e)}{\leq} (1 - \eta_{t+1}^f)^2 \mathbb{E}\|e_t^f\|^2 + 2(\eta_{t+1}^f)^2 \sigma_f^2$$
$$+ 2(1 - \eta_t^f)^2 \mathbb{E}\|(\bar{\nabla} f(x_{t+1}, y_{t+1}; \xi_{t+1}) - \bar{\nabla} f(x_{t+1}, y_{t+1}) - B_{t+1})$$
$$- (\bar{\nabla} f(x_t, y_t; \xi_{t+1}) - \bar{\nabla} f(x_t, y_t) - B_t)\|^2 \tag{33}$$

where equality $(a)$ uses the definition of the recursive gradient estimator (14); $(b)$ results from the definition $e_t^f := h_t^f - \bar{\nabla} f(x_t, y_t) - B_t$; $(c)$ follows from the fact that conditioned on $\mathcal{F}_{t+1} = \sigma\{y_0, x_0, \ldots, y_t, x_t, y_{t+1}, x_{t+1}\}$

$$\mathbb{E}\Big\langle e_t^f, (\bar{\nabla} f(x_{t+1}, y_{t+1}; \xi_{t+1}) - \bar{\nabla} f(x_{t+1}, y_{t+1}) - B_{t+1})$$
$$- (1 - \eta_{t+1}^f)\big((\bar{\nabla} f(x_t, y_t; \xi_{t+1}) - \bar{\nabla} f(x_t, y_t) - B_t)\big)\Big\rangle$$
$$\mathbb{E}\Big\langle e_t^f, \mathbb{E}\big[(\bar{\nabla} f(x_{t+1}, y_{t+1}; \xi_{t+1}) - \bar{\nabla} f(x_{t+1}, y_{t+1}) - B_{t+1})$$
$$\underbrace{- (1 - \eta_{t+1}^f)\big((\bar{\nabla} f(x_t, y_t; \xi_{t+1}) - \bar{\nabla} f(x_t, y_t) - B_t)\big)|\mathcal{F}_{t+1}\big]}_{=0}\Big\rangle = 0,$$

which follows from the fact that the second term in the inner product above is zero mean as a consequence of Assumption 4-(i) and inequality $(d)$ utilizes (24); and $(e)$ results from Assumption 4-(i).

Next, we bound the last term of (33) above

$$2(1-\eta_{t+1}^f)^2\mathbb{E}\big\|\big(\bar{\nabla}f(x_{t+1},y_{t+1};\xi_{t+1})-\bar{\nabla}f(x_t,y_t;\xi_{t+1})\big)$$
$$-\Big(\big(\bar{\nabla}f(x_{t+1},y_{t+1})+B_{t+1}\big)-\big(\bar{\nabla}f(x_t,y_t)+B_t\big)\Big)\big\|^2$$

$$\overset{(a)}{\leq} 2(1-\eta_{t+1}^f)^2\mathbb{E}\big\|\bar{\nabla}f(x_{t+1},y_{t+1};\xi_{t+1})-\bar{\nabla}f(x_t,y_t;\xi_{t+1})\big\|^2$$

$$\overset{(b)}{\leq} 4(1-\eta_{t+1}^f)^2\mathbb{E}\big\|\bar{\nabla}f(x_{t+1},y_{t+1};\xi_{t+1})-\bar{\nabla}f(x_t,y_{t+1};\xi_{t+1})\big\|^2$$
$$+4(1-\eta_{t+1}^f)^2\mathbb{E}\big\|\bar{\nabla}f(x_t,y_{t+1};\xi_{t+1})-\bar{\nabla}f(x_t,y_t;\xi_{t+1})\big\|^2$$

$$\overset{(c)}{\leq} 4(1-\eta_{t+1}^f)^2 L_K^2\mathbb{E}\|x_{t+1}-x_t\|^2+4(1-\eta_{t+1}^f)^2 L_K^2\mathbb{E}\|y_{t+1}-y_t\|^2$$

$$\overset{(d)}{\leq} 4(1-\eta_{t+1}^f)^2 L_K^2\alpha_t^2\mathbb{E}\|h_t^f\|^2+4(1-\eta_{t+1}^f)^2 L_K^2\beta_t^2\mathbb{E}\|h_t^g\|^2,$$

$$\overset{(e)}{\leq} 4(1-\eta_{t+1}^f)^2 L_K^2\alpha_t^2\mathbb{E}\|h_t^f\|^2+8(1-\eta_{t+1}^f)^2 L_K^2\beta_t^2\mathbb{E}\|e_t^g\|^2$$
$$+8(1-\eta_{t+1}^f)^2 L_K^2\beta_t^2\mathbb{E}\|\nabla_y g(x_t,y_t)\|^2, \tag{34}$$

where $(a)$ follows from the mean variance inequality: For a random variable $Z$ we have $\mathbb{E}\|Z-\mathbb{E}[Z]\|^2\leq\mathbb{E}\|Z\|^2$ with $Z$ defined as $Z\coloneqq\bar{\nabla}f(x_{t+1},y_{t+1};\xi_{t+1})-\bar{\nabla}f(x_t,y_t;\xi_{t+1})$; $(b)$ again uses (24); $(c)$ follows from Lemma C.2; inequality $(d)$ uses Steps 5 and 7 of Algorithm 1; finally, $(e)$ utilizes (24) and the definition of $e_t^g$.

Finally, substituting (34) in (33), we get the statement of the lemma.

Therefore, the lemma is proved. $\qquad\square$

## D.4 Descent in the gradient estimation error of the inner function

We consider the descent on the gradient estimation error of the inner function.

**Lemma D.4.** *Define $e_t^g\coloneqq h_t^g-\nabla_y g(x_t,y_t)$. Then the iterates generated from Algorithm 1 satisfy*

$$\mathbb{E}\|e_{t+1}^g\|^2\leq\Big((1-\eta_{t+1}^g)^2+8(1-\eta_{t+1}^g)^2 L_g^2\beta_t^2\Big)\mathbb{E}\|e_t^g\|^2+2(\eta_{t+1}^g)^2\sigma_g^2$$
$$+4(1-\eta_{t+1}^g)^2 L_g^2\alpha_t^2\mathbb{E}\|h_t^f\|^2+8(1-\eta_{t+1}^g)^2 L_g^2\beta_t^2\mathbb{E}\|\nabla_y g(x_t,y_t)\|^2$$

*for all $t\in\{0,1,\cdots,T-1\}$, where the expectation is taken w.r.t. the stochasticity of the algorithm.*

*Proof.* From the definition of $e_t^g$ we have

$$\mathbb{E}\|e_{t+1}^g\|^2 = \mathbb{E}\|h_{t+1}^g - \nabla_y g(x_{t+1}, y_{t+1})\|^2$$

$$\overset{(a)}{=} \mathbb{E}\|\nabla_y g(x_{t+1}, y_{t+1}, \zeta_{t+1}) + (1 - \eta_{t+1}^g)(h_t^g - \nabla_y g(x_t, y_t; \zeta_{t+1})) - \nabla_y g(x_{t+1}, y_{t+1})\|^2$$

$$\overset{(b)}{=} \mathbb{E}\|(1 - \eta_{t+1}^g)e_t^g + (\nabla_y g(x_{t+1}, y_{t+1}, \zeta_{t+1}) - \nabla_y g(x_{t+1}, y_{t+1}))$$
$$- (1 - \eta_{t+1}^g)(\nabla_y g(x_t, y_t; \zeta_{t+1}) - \nabla_y g(x_t, y_t))\|^2$$

$$\overset{(c)}{=} (1 - \eta_{t+1}^g)^2 \mathbb{E}\|e_t^g\|^2$$
$$+ \mathbb{E}\|\nabla_y g(x_{t+1}, y_{t+1}, \zeta_{t+1}) - \nabla_y g(x_{t+1}, y_{t+1}) - (1 - \eta_{t+1}^g)(\nabla_y g(x_t, y_t; \zeta_{t+1}) - \nabla_y g(x_t, y_t))\|^2$$

$$\overset{(d)}{\leq} (1 - \eta_{t+1}^g)^2 \mathbb{E}\|e_t^g\|^2 + 2(\eta_{t+1}^g)^2 \sigma_g^2 + 2(1 - \eta_{t+1}^g)^2 \mathbb{E}\|\nabla_y g(x_{t+1}, y_{t+1}, \zeta_{t+1}) - \nabla_y g(x_t, y_t; \zeta_{t+1})\|^2$$

$$\overset{(e)}{\leq} (1 - \eta_{t+1}^g)^2 \mathbb{E}\|e_t^g\|^2 + 2(\eta_{t+1}^g)^2 \sigma_g^2$$
$$+ 4(1 - \eta_{t+1}^g)^2 \mathbb{E}\|\nabla_y g(x_{t+1}, y_{t+1}, \zeta_{t+1}) - \nabla_y g(x_t, y_{t+1}; \zeta_{t+1})\|^2$$
$$+ 4(1 - \eta_{t+1}^g)^2 \mathbb{E}\|\nabla_y g(x_t, y_{t+1}, \zeta_{t+1}) - \nabla_y g(x_t, y_t; \zeta_{t+1})\|^2$$

$$\overset{(f)}{\leq} (1 - \eta_{t+1}^g)^2 \mathbb{E}\|e_t^g\|^2 + 2(\eta_{t+1}^g)^2 \sigma_g^2 + 4(1 - \eta_{t+1}^g)^2 L_g^2 \mathbb{E}\|x_{t+1} - x_t\|^2 + 4(1 - \eta_{t+1}^g)^2 L_g^2 \mathbb{E}\|y_{t+1} - y_t\|^2$$

$$\overset{(g)}{\leq} (1 - \eta_{t+1}^g)^2 \mathbb{E}\|e_t^g\|^2 + 2(\eta_{t+1}^g)^2 \sigma_g^2 + 4(1 - \eta_{t+1}^g)^2 L_g^2 \alpha_t^2 \mathbb{E}\|h_t^f\|^2 + 4(1 - \eta_{t+1}^g)^2 L_g^2 \beta_t^2 \mathbb{E}\|h_t^g\|^2$$

$$\overset{(h)}{\leq} (1 - \eta_{t+1}^g)^2 \mathbb{E}\|e_t^g\|^2 + 2(\eta_{t+1}^g)^2 \sigma_g^2 + 4(1 - \eta_{t+1}^g)^2 L_g^2 \alpha_t^2 \mathbb{E}\|h_t^f\|^2$$
$$+ 8(1 - \eta_{t+1}^g)^2 L_g^2 \beta_t^2 \mathbb{E}\|e_t^g\|^2 + 8(1 - \eta_{t+1}^g)^2 L_g^2 \beta_t^2 \mathbb{E}\|\nabla_y g(x_t, y_t)\|^2$$

$$\leq \Big((1 - \eta_{t+1}^g)^2 + 8(1 - \eta_{t+1}^g)^2 L_g^2 \beta_t^2\Big)\mathbb{E}\|e_t^g\|^2 + 2(\eta_{t+1}^g)^2 \sigma_g^2 + 4(1 - \eta_{t+1}^g)^2 L_g^2 \alpha_t^2 \mathbb{E}\|h_t^f\|^2$$
$$+ 8(1 - \eta_{t+1}^g)^2 L_g^2 \beta_t^2 \mathbb{E}\|\nabla_y g(x_t, y_t)\|^2,$$

where equality $(a)$ uses the definition of hybrid gradient estimator (13); $(b)$ uses the definition of $e_t^g$; $(c)$ uses the fact that conditioned on $\mathcal{F}_{t+1} = \sigma\{y_0, x_0, \ldots, y_t, x_t, y_{t+1}, x_{t+1}\}$

$$\mathbb{E}\langle e_t^g, (\nabla_y g(x_{t+1}, y_{t+1}, \zeta_{t+1}) - \nabla_y g(x_{t+1}, y_{t+1})) - (1 - \eta_{t+1}^g)(\nabla_y g(x_t, y_t; \zeta_{t+1}) - \nabla_y g(x_t, y_t))\rangle$$
$$= \mathbb{E}\langle e_t^g, \underbrace{\mathbb{E}[(\nabla_y g(x_{t+1}, y_{t+1}, \zeta_{t+1}) - \nabla_y g(x_{t+1}, y_{t+1})) - (1 - \eta_{t+1}^g)(\nabla_y g(x_t, y_t; \zeta_{t+1}) - \nabla_y g(x_t, y_t))|\mathcal{F}_{t+1}]}_{=0}\rangle$$
$$= 0.$$

Inequality $(d)$ results from the application of (24) and Assumption 4-(ii); $(e)$ again uses (24); $(f)$ utilizes Assumption 2; $(g)$ follows from Steps 5 and 7 of Algorithm 1 and finally, $(h)$ follows from the application of (24) and the definition of $e_t^g$.

Therefore, the lemma is proved. $\qquad\square$

### D.5 Descent in the potential function

Let us define the potential function as:

$$V_t := \ell(x_t) + \frac{2L}{3\sqrt{2}L_y}\|y_t - y^*(x_t)\|^2 + \frac{1}{\bar{c}_{\eta_f}}\frac{\|e_t^f\|^2}{\alpha_{t-1}} + \frac{1}{\bar{c}_{\eta_g}}\frac{\|e_t^g\|^2}{\alpha_{t-1}} \tag{35}$$

where we define

$$\bar{c}_{\eta_f} := \max\left\{36L_K^2, \frac{4L_K^2 L_{\mu_g}(\mu_g + L_g)c_\beta^2}{L^2}\right\} \quad \text{and} \quad \bar{c}_{\eta_g} := \max\left\{36L_g^2, \frac{4L_g^2 L_{\mu_g}(\mu_g + L_g)c_\beta^2}{L^2}\right\}. \tag{36}$$

with $L_{\mu_g}$ defined as $L_{\mu_g} := \frac{\mu_g L_g}{\mu_g + L_g}$.

Next, we quantify the expected descent in the potential function $\mathbb{E}[V_{t+1} - V_t]$.

**Lemma D.5.** *Consider $V_t$ defined in (35). Suppose the parameters of Algorithm 1 are chosen as*

$$\alpha_t := \frac{1}{(w+t)^{1/3}}, \ \beta_t := c_\beta \alpha_t, \ \eta_{t+1}^f := c_{\eta_f} \alpha_t^2, \ and \ \eta_{t+1}^g := c_{\eta_g} \alpha_t^2 \ for \ all \ t \in \{0, 1, \ldots, T-1\}.$$

*with*

$$c_\beta := \frac{6\sqrt{2}L_y L}{L_{\mu_g}}, \ c_{\eta_f} := \frac{1}{3L_f} + \bar{c}_{\eta_f} \ and \ c_{\eta_g} := \frac{1}{3L_f} + 8L_g^2 c_\beta^2 + \left[\frac{8L^2}{L_{\mu_g}^2} + \frac{2L^2}{L_{\mu_g}(\mu_g + L_g)}\right]\bar{c}_{\eta_g},$$

*where $L_{\mu_g} := \frac{\mu_g L_g}{\mu_g + L_g}$ and*

$$\bar{c}_{\eta_f} = \max\left\{36L_K^2, \frac{4L_K^2 L_{\mu_g}(\mu_g + L_g)c_\beta^2}{L^2}\right\} \quad and \quad \bar{c}_{\eta_g} = \max\left\{36L_g^2, \frac{4L_g^2 L_{\mu_g}(\mu_g + L_g)c_\beta^2}{L^2}\right\}.$$

*and the parameters*

$$\gamma_t := \frac{\beta_t L_{\mu_g}/2}{1 - \beta_t L_{\mu_g}} \quad and \quad \delta_t := \frac{\beta_t L_{\mu_g}}{1 - 2\beta_t L_{\mu_g}}.$$

*Then the iterates generated by Algorithm 1 when the outer problem is non-convex satisfy:*

$$\mathbb{E}[V_{t+1} - V_t] \le -\frac{\alpha_t}{2}\mathbb{E}\|\nabla\ell(x_t)\|^2 + 2\alpha_t\|B_t\|^2 + \frac{2(\eta_{t+1}^f)^2}{\bar{c}_{\eta_f}\alpha_t}\sigma_f^2 + \frac{2(\eta_{t+1}^g)^2}{\bar{c}_{\eta_g}\alpha_t}\sigma_g^2.$$

*for all $t \in \{0, 1, \ldots, T-1\}$*

*Proof.* We have from Lemma D.2

$$
\begin{aligned}
\mathbb{E}\|y_{t+1} - y^*(x_{t+1})\|^2 - \mathbb{E}\|y_t - y^*(x_t)\|^2 &\le \left[(1+\gamma_t)(1+\delta_t)\left(1 - 2\beta_t\frac{\mu_g L_g}{\mu_g + L_g}\right) - 1\right]\mathbb{E}\|y_t - y^*(x_t)\|^2 \\
&\quad - (1+\gamma_t)(1+\delta_t)\left(\frac{2\beta_t}{\mu_g + L_g} - \beta_t^2\right)\mathbb{E}\|\nabla_y g(x_t, y_t)\| \\
&\quad + (1+\gamma_t)\left(1 + \frac{1}{\delta_t}\right)\beta_t^2\mathbb{E}\|e_t^g\|^2 + \left(1 + \frac{1}{\gamma_t}\right)L_y^2\alpha_t^2\mathbb{E}\|h_t^f\|^2.
\end{aligned}
$$

$$(37)$$

Let us consider coefficient of the first term of (37) above, choosing $\gamma_t$ and $\delta_t$ such that we have

$$(1+\gamma_t)(1+\delta_t)(1 - 2\beta_t L_{\mu_g}) = 1 - \frac{\beta_t L_{\mu_g}}{2} \tag{38}$$

where we define $L_{\mu_g} := \frac{\mu_g L_g}{\mu_g + L_g}$. First we choose $\gamma_t$ such that we have

$$(1+\delta_t)(1 - 2\beta_t L_{\mu_g}) = 1 - \beta_t L_{\mu_g} \quad \Rightarrow \quad 1 + \delta_t = \frac{1 - \beta_t L_{\mu_g}}{1 - 2\beta_t L_{\mu_g}} \quad \Rightarrow \quad \delta_t = \frac{\beta_t L_{\mu_g}}{1 - 2\beta_t L_{\mu_g}}$$

Moreover, this implies that we have:

$$1 + \frac{1}{\delta_t} = 1 + \frac{1 - 2\beta_t L_{\mu_g}}{\beta_t L_{\mu_g}} \le \frac{1}{\beta_t L_{\mu_g}}.$$

Using the definition of $\delta_t$ in (38) we

$$(1+\gamma_t)(1 - \beta_t L_{\mu_g}) = 1 - \frac{\beta_t L_{\mu_g}}{2} \quad \Rightarrow \quad 1 + \gamma_t = \frac{1 - \frac{\beta_t L_{\mu_g}}{2}}{1 - \beta_t L_{\mu_g}} \quad \Rightarrow \quad \gamma_t = \frac{\beta_t L_{\mu_g}/2}{1 - \beta_t L_{\mu_g}}$$

Moreover, this implies that we have:

$$1 + \frac{1}{\gamma_t} = 1 + \frac{1 - \beta_t L_{\mu_g}}{\beta_t L_{\mu_g}/2} \le \frac{2}{\beta_t L_{\mu_g}}.$$

Substituting the above bounds in (37), we get

$$\mathbb{E}\|y_{t+1} - y^*(x_{t+1})\|^2 - \mathbb{E}\|y_t - y^*(x_t)\|^2 \leq -\frac{\beta_t L_{\mu_g}}{2}\mathbb{E}\|y_t - y^*(x_t)\|^2 - \left(\frac{2\beta_t}{\mu_g + L_g} - \beta_t^2\right)\mathbb{E}\|\nabla_y g(x_t, y_t)\|$$
$$+ \frac{2}{\beta_t L_{\mu_g}}\beta_t^2\mathbb{E}\|e_t^g\|^2 + \frac{2}{\beta_t L_{\mu_g}}L_y^2\alpha_t^2\mathbb{E}\|h_t^f\|^2.$$

Choosing $\beta_t \leq \frac{1}{\mu_g + L_g}$ we get

$$\mathbb{E}\|y_{t+1} - y^*(x_{t+1})\|^2 - \mathbb{E}\|y_t - y^*(x_t)\|^2 \leq -\frac{\beta_t L_{\mu_g}}{2}\mathbb{E}\|y_t - y^*(x_t)\|^2 - \frac{\beta_t}{\mu_g + L_g}\mathbb{E}\|\nabla_y g(x_t, y_t)\|$$
$$+ \frac{2}{\beta_t L_{\mu_g}}\beta_t^2\mathbb{E}\|e_t^g\|^2 + \frac{2}{\beta_t L_{\mu_g}}L_y^2\alpha_t^2\mathbb{E}\|h_t^f\|^2.$$

Using the definition of $\beta_t = c_\beta \alpha_t$ and multiplying both sides by $\frac{4L^2}{c_\beta L_{\mu_g}}$ we get

$$\frac{4L^2}{c_\beta L_{\mu_g}}\mathbb{E}\left[\|y_{t+1} - y^*(x_{t+1})\|^2 - \|y_t - y^*(x_t)\|^2\right] \leq -2\alpha_t L^2\mathbb{E}\|y_t - y^*(x_t)\|^2 - \frac{4L^2\alpha_t}{L_{\mu_g}(\mu_g + L_g)}\mathbb{E}\|\nabla_y g(x_t, y_t)\|$$
$$+ \frac{8L^2\alpha_t}{L_{\mu_g}^2}\mathbb{E}\|e_t^g\|^2 + \frac{8L_y^2 L^2\alpha_t}{c_\beta^2 L_{\mu_g}^2}\mathbb{E}\|h_t^f\|^2.$$

Finally, choosing $c_\beta = \frac{6\sqrt{2}L_y L}{L_{\mu_g}}$ such that $\frac{8L_y^2 L^2}{c_\beta^2 L_{\mu_g}^2} = \frac{1}{9}$

$$\frac{2L}{3\sqrt{2}L_y}\mathbb{E}\left[\|y_{t+1} - y^*(x_{t+1})\|^2 - \|y_t - y^*(x_t)\|^2\right] \leq -2\alpha_t L^2\mathbb{E}\|y_t - y^*(x_t)\|^2 - \frac{4L^2\alpha_t}{L_{\mu_g}(\mu_g + L_g)}\mathbb{E}\|\nabla_y g(x_t, y_t)\|$$
$$+ \frac{8L^2\alpha_t}{L_{\mu_g}^2}\mathbb{E}\|e_t^g\|^2 + \frac{\alpha_t}{9}\mathbb{E}\|h_t^f\|^2. \tag{39}$$

Next, we have from Lemma D.3

$$\frac{\mathbb{E}\|e_{t+1}^f\|^2}{\alpha_t} - \frac{\mathbb{E}\|e_{t+1}^f\|^2}{\alpha_{t-1}} \leq \left[\frac{(1 - \eta_{t+1}^f)^2}{\alpha_t} - \frac{1}{\alpha_{t-1}}\right]\mathbb{E}\|e_t^f\|^2 + \frac{2(\eta_{t+1}^f)^2}{\alpha_t}\sigma_f^2 + 4L_K^2\alpha_t\mathbb{E}\|h_t^f\|^2$$
$$+ \frac{8L_K^2\beta_t^2}{\alpha_t}\mathbb{E}\|e_t^g\|^2 + \frac{8L_K^2\beta_t^2}{\alpha_t}\mathbb{E}\|\nabla_y g(x_t, y_t)\|^2, \tag{40}$$

where we have utilized the fact that $0 < 1 - \eta_t < 1$ for all $t \in \{0, 1, \ldots, T - 1\}$. Now we consider the coefficient of the first term on the right hand side of (40), we have

$$\frac{(1 - \eta_{t+1}^f)^2}{\alpha_t} - \frac{1}{\alpha_{t-1}} \leq \frac{1}{\alpha_t} - \frac{\eta_{t+1}^f}{\alpha_t} - \frac{1}{\alpha_{t-1}}. \tag{41}$$

Using the definition of $\alpha_t$ we have

$$\frac{1}{\alpha_t} - \frac{1}{\alpha_{t-1}} = (w + t)^{1/3} - (w + t - 1)^{1/3}] \overset{(a)}{\leq} \frac{1}{3(w + t - 1)^{2/3}} \overset{(b)}{\leq} \frac{1}{3(w/2 + t)^{2/3}}$$
$$= \frac{2^{2/3}}{3(w + 2t)^{2/3}} \leq \frac{2^{2/3}}{3(w + t)^{2/3}} \overset{(c)}{\leq} \frac{2^{2/3}}{3}\alpha_t^2 \overset{(d)}{\leq} \frac{\alpha_t}{3L_f},$$

where $(a)$ follows from $(x + y)^{1/3} - x^{1/3} \leq y/(3x^{2/3})$; $(b)$ results from the fact that we choose $w \geq 2$ hence $1 \leq w/2$; $(c)$ results from the definition of $\alpha_t$ and $(d)$ uses the fact that we choose $\alpha_t \leq 1/3L_f$. Substituting in (41) and using $\eta_{t+1}^f = c_{\eta_f}\alpha_t^2$, we get

$$\frac{(1 - \eta_{t+1}^f)^2}{\alpha_t} - \frac{1}{\alpha_{t-1}} \leq \frac{\alpha_t}{3L_f} - c_{\eta_f}\alpha_t \leq -\bar{c}_{\eta_f}\alpha_t,$$

which follows from the choice

$$c_{\eta_f} = \frac{1}{3L_f} + \bar{c}_{\eta_f} \quad \text{with} \quad \bar{c}_{\eta_f} = \max\left\{36L_K^2, \frac{4L_K^2 L_{\mu_g}(\mu_g + L_g)c_\beta^2}{L^2}\right\}.$$

Substiuting in (40)

$$\frac{1}{\bar{c}_{\eta_f}}\mathbb{E}\left[\frac{\|e_{t+1}^f\|^2}{\alpha_t} - \frac{\|e_{t+1}^f\|^2}{\alpha_{t-1}}\right] \leq -\alpha_t\mathbb{E}\|e_t^f\|^2 + \frac{2(\eta_{t+1}^f)^2}{\bar{c}_{\eta_f}\alpha_t}\sigma_f^2 + \frac{\alpha_t}{9}\mathbb{E}\|h_t^f\|^2 + \frac{2L^2}{L_{\mu_g}(\mu_g + L_g)}\alpha_t\mathbb{E}\|e_t^g\|^2$$
$$+ \frac{2L^2}{L_{\mu_g}(\mu_g + L_g)}\alpha_t\mathbb{E}\|\nabla_y g(x_t, y_t)\|^2,$$
(42)

Next, from Lemma D.4, we have

$$\frac{\mathbb{E}\|e_{t+1}^g\|^2}{\alpha_t} - \frac{\mathbb{E}\|e_t^g\|^2}{\alpha_{t-1}} \leq \left[\frac{(1 - \eta_{t+1}^g)^2 + 8(1 - \eta_{t+1}^g)^2 L_g^2\beta_t^2}{\alpha_t} - \frac{1}{\alpha_{t-1}}\right]\mathbb{E}\|e_t^g\|^2 + \frac{2(\eta_{t+1}^g)^2}{\alpha_t}\sigma_g^2$$
$$+ 4L_g^2\alpha_t\mathbb{E}\|h_t^f\|^2 + \frac{8L_g^2\beta_t^2}{\alpha_t}\mathbb{E}\|\nabla_y g(x_t, y_t)\|^2$$
(43)

where we have utilized the fact that $0 < 1 - \eta_t^g \leq 1$ for all $t \in \{0, 1, \ldots, T-1\}$. Let us consider the coefficient of the first term on the right hand side of (43) we have

$$\frac{(1 - \eta_{t+1}^g)^2 + 8(1 - \eta_{t+1}^g)^2 L_g^2\beta_t^2}{\alpha_t} - \frac{1}{\alpha_{t-1}} \leq \frac{(1 - \eta_{t+1}^g)}{\alpha_t}\left(1 + 8L_g^2\beta_t^2\right) - \frac{1}{\alpha_{t-1}}$$
$$= \frac{1}{\alpha_t} - \frac{1}{\alpha_{t-1}} + \frac{8L_g^2\beta_t^2}{\alpha_t} - c_{\eta_g}\alpha_t(1 + 8L_g^2\beta_t^2),$$

using the fact that from earlier we have $\frac{1}{\alpha_t} - \frac{1}{\alpha_{t-1}} \leq \frac{\alpha_t}{3L_f}$ and the definition of $\beta_t = c_\beta\alpha_t$, we have

$$\frac{(1 - \eta_{t+1}^g)^2 + 8(1 - \eta_{t+1}^g)^2 L_g^2\beta_t^2}{\alpha_t} - \frac{1}{\alpha_{t-1}} \leq \frac{\alpha_t}{3L_f} + 8L_g^2 c_\beta^2\alpha_t - c_{\eta_g}\alpha_t,$$

Next choosing $c_{\eta_g}$ as

$$c_{\eta_g} = \frac{1}{3L_f} + 8L_g^2 c_\beta^2 + \left[\frac{8L^2}{L_{\mu_g}^2} + \frac{2L^2}{L_{\mu_g}(\mu_g + L_g)}\right]\bar{c}_{\eta_g} \quad \text{with} \quad \bar{c}_{\eta_g} = \max\left\{36L_g^2, \frac{4L_g^2 L_{\mu_g}(\mu_g + L_g)c_\beta^2}{L^2}\right\}.$$

Therefore, we get

$$\frac{(1 - \eta_{t+1}^g)^2 + 8(1 - \eta_{t+1}^g)^2 L_g^2\beta_t^2}{\alpha_t} - \frac{1}{\alpha_{t-1}} \leq -\left[\frac{8L^2}{L_{\mu_g}^2} + \frac{2L^2}{L_{\mu_g}(\mu_g + L_g)}\right]\bar{c}_{\eta_g}\alpha_t,$$

Finally, replacing in (43) we get

$$\frac{1}{\bar{c}_{\eta_g}}\mathbb{E}\left[\frac{\|e_{t+1}^g\|^2}{\alpha_t} - \frac{\|e_t^g\|^2}{\alpha_{t-1}}\right] \leq -\left[\frac{8L^2}{L_{\mu_g}^2} + \frac{2L^2}{L_{\mu_g}(\mu_g + L_g)}\right]\alpha_t\mathbb{E}\|e_t^g\|^2 + \frac{2(\eta_{t+1}^g)^2}{\bar{c}_{\eta_g}\alpha_t}\sigma_g^2$$
$$+ \frac{\alpha_t}{9}\mathbb{E}\|h_t^f\|^2 + \frac{2L^2}{L_{\mu_g}(\mu_g + L_g)}\alpha_t\mathbb{E}\|\nabla_y g(x_t, y_t)\|^2$$
(44)

Finally, adding (39), (42), (44) and the result of Lemma D.1 with $\alpha_t \leq 1/3L_f$, we get

$$\mathbb{E}[V_{t+1} - V_t] \leq -\frac{\alpha_t}{2}\mathbb{E}\|\nabla\ell(x_t)\|^2 + 2\alpha_t\|B_t\|^2 + \frac{2(\eta_{t+1}^f)^2}{\bar{c}_{\eta_f}\alpha_t}\sigma_f^2 + \frac{2(\eta_{t+1}^g)^2}{\bar{c}_{\eta_g}\alpha_t}\sigma_g^2.$$

Therefore, we have the statement of the Lemma.

### D.6 Proof of Theorem 3.2

Summing the result of Lemma D.5 for $t = 0$ to $T - 1$, dividing by $T$ on both sides and using the definition $\eta^f_{t+1} := c_{\eta_f} \alpha_t^2$ and $\eta^g_{t+1} := c_{\eta_g} \alpha_t^2$ we get

$$\frac{\mathbb{E}[V_T - V_0]}{T} \leq -\frac{1}{T} \sum_{t=0}^{T-1} \frac{\alpha_t}{2} \mathbb{E}\|\nabla \ell(x_t)\|^2 + \frac{2}{T} \sum_{t=0}^{T} \alpha_t \|B_t\|^2 + \frac{2c_{\eta_f}^2 \sigma_f^2}{\bar{c}_{\eta_f}} \sum_{t=0}^{T-1} \alpha_t^3 + \frac{2c_{\eta_g}^2 \sigma_g^2}{\bar{c}_{\eta_g}} \sum_{t=0}^{T-1} \alpha_t^3.$$
(45)

Next considering $\sum_{t=0}^{T-1} \alpha_t$ in the last two terms on the right hand side of (45), we have from the definition of $\alpha_t$ that

$$\sum_{t=0}^{T-1} \alpha_t^3 = \sum_{t=0}^{T-1} \frac{1}{w+t} \overset{(a)}{\leq} \sum_{t=0}^{T-1} \frac{1}{1+t} \leq \log(T+1)$$

where inequality $(a)$ results from the fact that we choose $w \geq 1$. Substituting the above in (45) we get

$$\frac{\mathbb{E}[V_T - V_0]}{T} \leq -\frac{1}{T} \sum_{t=0}^{T-1} \frac{\alpha_t}{2} \mathbb{E}\|\nabla \ell(x_t)\|^2 + \frac{2}{T} \sum_{t=0}^{T} \alpha_t \|B_t\|^2 + \frac{2c_{\eta_f}^2}{\bar{c}_{\eta_f}} \frac{\log(T+1)}{T} \sigma_f^2 + \frac{2c_{\eta_g}^2}{\bar{c}_{\eta_g}} \frac{\log(T+1)}{T} \sigma_g^2$$

Rearranging the terms we get

$$\frac{1}{T} \sum_{t=0}^{T-1} \frac{\alpha_t}{2} \mathbb{E}\|\nabla \ell(x_t)\|^2 \leq \frac{\mathbb{E}[V_0 - \ell^*]}{T} + \frac{2}{T} \sum_{t=0}^{T} \alpha_t \|B_t\|^2 + \frac{2c_{\eta_f}^2}{\bar{c}_{\eta_f}} \frac{\log(T+1)}{T} \sigma_f^2 + \frac{2c_{\eta_g}^2}{\bar{c}_{\eta_g}} \frac{\log(T+1)}{T} \sigma_g^2$$

Using the fact that $\alpha_t$ is decreasing in $t$ we have $\alpha_T \leq \alpha_t$ for all $t \in \{0, 1, \dots, T-1\}$ and multiplying by $2/\alpha_T$ on both sides we get

$$\frac{1}{T} \sum_{t=0}^{T-1} \mathbb{E}\|\nabla \ell(x_t)\|^2 \leq \frac{2\mathbb{E}[V_0 - \ell^*]}{\alpha_T T} + \frac{4}{\alpha_T T} \sum_{t=0}^{T} \alpha_t \|B_t\|^2 + \frac{4c_{\eta_f}^2}{\bar{c}_{\eta_f}} \frac{\log(T+1)}{\alpha_T T} \sigma_f^2 + \frac{4c_{\eta_g}^2}{\bar{c}_{\eta_g}} \frac{\log(T+1)}{\alpha_T T} \sigma_g^2$$

Finally, we have from the definition of the Potential function

$$\mathbb{E}[V_0] := \mathbb{E}\left[\ell(x_0) + \frac{2L}{3\sqrt{2}L_y} \|y_0 - y^*(x_0)\|^2 + \frac{1}{\bar{c}_{\eta_f}} \frac{\|e_0^f\|^2}{\alpha_{-1}} + \frac{1}{\bar{c}_{\eta_g}} \frac{\|e_0^g\|^2}{\alpha_{-1}}\right]$$

$$\leq \ell(x_0) + \frac{2L}{3\sqrt{2}L_y} \|y_0 - y^*(x_0)\|^2 + \frac{\sigma_f^2}{\bar{c}_{\eta_f} \alpha_{-1}} + \frac{\sigma_g^2}{\bar{c}_{\eta_g} \alpha_{-1}},$$

which follows from the assumption and the definition of $h_t^f$ and $h_t^g$. Therefore, we have

$$\frac{1}{T} \sum_{t=0}^{T-1} \|\nabla \ell(x_t)\|^2 \leq \frac{2(\ell(x_0) - \ell^*)}{\alpha_T T} + \frac{4L}{3\sqrt{2}L_y} \frac{\|y_0 - y^*(x_0)\|^2}{\alpha_T T} + \frac{2}{\bar{c}_{\eta_f} \alpha_{-1}} \frac{\sigma_f^2}{\alpha_T T} + \frac{2}{\bar{c}_{\eta_g} \alpha_{-1}} \frac{\sigma_g^2}{\alpha_T T}$$

$$+ \frac{4}{\alpha_T T} \sum_{t=0}^{T} \alpha_t \|B_t\|^2 + \frac{4c_{\eta_f}^2}{\bar{c}_{\eta_f}} \frac{\log(T+1)}{\alpha_T T} \sigma_f^2 + \frac{4c_{\eta_g}^2}{\bar{c}_{\eta_g}} \frac{\log(T+1)}{\alpha_T T} \sigma_g^2$$

Finally, we have from the definition of $\alpha_T := 1/(w+T)^{1/3}$ and $\alpha_1 = \alpha_0$, moreover using the fact that for the choice of $K = (L_g/\mu_g) \log(C_{g_{xy}} C_{f_y} T/\mu_g)$ stochastic Hessian samples of $\nabla^2_{yy} g(x, y)$ we have $\|B_t\| = 1/T$, we get

$$\mathbb{E}\|\nabla \ell(x_a(T))\|^2 \leq \mathcal{O}\left(\frac{\ell(x_0) - \ell^*}{T^{2/3}}\right) + \mathcal{O}\left(\frac{\|y_0 - y^*(x_0)\|^2}{T^{2/3}}\right) + \tilde{\mathcal{O}}\left(\frac{\sigma_f^2}{T^{2/3}}\right) + \tilde{\mathcal{O}}\left(\frac{\sigma_g^2}{T^{2/3}}\right).$$

Hence, the theorem is proved. $\qquad \square$

# E   Proof of Theorem 3.3: strongly-convex outer objective

To prove Theorem 3.3, we utilize the descent results obtained for the proof of Theorem 3.2 in Appendix D. The proof follows similar structure as the proof of non-convex case. We first consider the descent achieved by the consecutive iterates generated by Algorithm 1 when the outer function is strongly-convex and smooth.

## E.1   Descent in the function value

**Lemma E.1.** *For strongly-convex and smooth $\ell(\cdot)$, with $e_t^f$ defined as: $e_t^f := h_t^f - \bar{\nabla} f(x_t, y_{t+1}) - B_t$, the consecutive iterates of Algorithm 1 satisfy:*

$$\mathbb{E}[\ell(x_{t+1}) - \ell^*] \leq \mathbb{E}\Big[(1 - \alpha_t \mu_f)\big(\ell(x_t) - \ell^*\big) - \frac{\alpha_t}{2}(1 - \alpha_t L_f)\|h_t^f\|^2 + \alpha_t \|e_t^f\|^2$$
$$+ 2\alpha_t L^2 \|y_t - y^*(x_t)\|^2 + 2\alpha_t \|B_t\|^2\Big],$$

*for all $t \in \{0, 1, \ldots, T-1\}$, where the expectation is w.r.t. the stochasticity of the algorithm.*

*Proof.* Note that from Lemma D.1 derived in Appendix D, we have

$$\mathbb{E}[\ell(x_{t+1})] \leq \mathbb{E}\Big[\ell(x_t) - \frac{\alpha_t}{2}\|\nabla\ell(x_t)\|^2 - \frac{\alpha_t}{2}(1 - \alpha_t L_f)\|h_t^f\|^2 + \alpha_t\|e_t^f\|^2 \tag{46}$$
$$+ 2\alpha_t L^2 \|y_t - y^*(x_t)\|^2 + 2\alpha_t \|B_t\|^2\Big].$$

Now using the fact that for a strongly convex function we have:

$$\|\nabla\ell(x)\|^2 \geq 2\mu_f(\ell(x) - \ell^*) \quad \text{for all} \quad x \in \mathbb{R}^{d_{\mathsf{up}}},$$

substituting in (46), subtracting $\ell^*$ from both sides and rearranging the terms yields the statement of the Lemma. $\qquad\square$

## E.2   Descent in the iterates of the lower level problem

**Lemma E.2.** *The iterates of the inner problem generated according to Algorithm 1, satisfy*

$$\mathbb{E}\|y_{t+1} - y^*(x_{t+1})\|^2 \leq (1 + \gamma_t)\big(1 - 2\beta_t \mu_g + \beta_t^2 L_g^2\big)\mathbb{E}\|y_t - y^*(x_t)\|^2$$
$$+ \left(1 + \frac{1}{\gamma_t}\right)L_y^2 \alpha_t^2 \mathbb{E}\|h_t^f\|^2 + (1 + \gamma_t)\beta_t^2 \sigma_g^2.$$

*for all $t \in \{0, \ldots, T-1\}$ with some $\gamma_t > 0$, where the expectation is w.r.t. the stochasticity of the algorithm.*

*Proof.* Consider the term $\mathbb{E}\|y_{t+1} - y^*(x_{t+1})\|^2$, we have

$$\mathbb{E}\|y_{t+1} - y^*(x_{t+1})\|^2 \overset{(a)}{\leq} (1 + \gamma_t)\mathbb{E}\|y_{t+1} - y^*(x_t)\|^2 + \left(1 + \frac{1}{\gamma_t}\right)\mathbb{E}\|y^*(x_{t+1}) - y^*(x_t)\|^2$$

$$\overset{(b)}{\leq} (1 + \gamma_t)\mathbb{E}\|y_t - \beta_t h_t^g - y^*(x_t)\|^2 + \left(1 + \frac{1}{\gamma_t}\right)L_y^2 \mathbb{E}\|x_{t+1} - x_t\|^2$$

$$\overset{(c)}{\leq} (1 + \gamma_t)\mathbb{E}\|y_t - \beta_t h_t^g - y^*(x_t)\|^2 + \left(1 + \frac{1}{\gamma_t}\right)L_y^2 \alpha_t^2 \mathbb{E}\|h_t^f\|^2 \tag{47}$$

where $(a)$ results from Young's inequality; $(b)$ uses Step 5 of Algorithm 1 and Lipschitzness of $y^*(\cdot)$ given in Lemma 2.2; and $(c)$ uses Step 7 of Algorithm 1.

Next, we consider the first term of (47) above:

$$\mathbb{E}\|y_t - \beta_t h_t^g - y^*(x_t)\|^2 = \mathbb{E}\|y_t - y^*(x_t)\|^2 + \beta_t^2 \mathbb{E}\|h_t^g\|^2 - \beta_t \mathbb{E}\langle y_t - y^*(x_t), h_t^g\rangle$$

$$\overset{(a)}{\leq} \mathbb{E}\|y_t - y^*(x_t)\|^2 + \beta_t^2 \mathbb{E}\|\nabla_y g(x_t, y_t)\|^2 + \beta_t^2 \mathbb{E}\|h_t^g - \nabla_y g(x_t, y_t)\|^2$$
$$- \beta_t \mathbb{E}\langle y_t - y^*(x_t), \nabla_y g(x_t, y_t)\rangle$$

$$\overset{(b)}{\leq} (1 - 2\mu_g \beta_t + \beta_t^2 L_g^2)\mathbb{E}\|y_t - y^*(x_t)\|^2 + + \beta_t^2 \sigma_g^2 \tag{48}$$

where $(a)$ utilizes the fact that for $\eta_t^g = 1$ we have $\mathbb{E}[h_t^g | \mathcal{F}_t] = \nabla_y g(x_t, y_t)$ and $(b)$ uses the fact that (1) $\nabla_y g(x, y^*(x)) = 0$ and the Lipschitzness of $\nabla_y g(x, \cdot)$ in Assumption 2-(ii); (2) Assumption 4-(ii); and (3) $g(x,y)$ is $\mu_g$-strongly convex w.r.t. $y$, we therefore have

$$\langle \nabla g_y(x, y_1) - \nabla g_y(x, y_2), y_1 - y_2 \rangle \geq \mu_g \|y_1 - y_2\|^2,$$

using $y_1 = y_t$ and $y_2 = y^*(x_t)$ yields inequality $(b)$. Finally, substituting (48) in (47) yields the statement of the lemma. $\qquad\square$

### E.3 Descent in the gradient estimation error

**Lemma E.3.** *Define $e_t^f := h_t^f - \bar{\nabla} f(x_t, y_t) - B_t$. Then the consecutive iterates of Algorithm 1 satisfy:*

$$\mathbb{E}\|e_{t+1}^f\|^2 \leq (1 - \eta_{t+1}^f)^2 \mathbb{E}\|e_t^f\|^2 + 2(\eta_{t+1}^f)^2 \sigma_f^2 + 4(1 - \eta_{t+1}^f)^2 L_K^2 \alpha_t^2 \mathbb{E}\|h_t^f\|^2$$
$$+ 8(1 - \eta_{t+1}^f)^2 L_K^2 \beta_t^2 \sigma_g^2 + 8(1 - \eta_{t+1}^f)^2 L_K^2 L_g^2 \beta_t^2 \mathbb{E}\|y_t - y^*(x_t)\|^2,$$

*for all $t \in \{0, \ldots, T - 1\}$, with $L_K$ defined in the statement of Lemma C.2. Here the expectation is taken w.r.t the stochasticity of the algorithm.*

*Proof.* From the statement of Lemma D.3, we have

$$\mathbb{E}\|e_{t+1}^f\|^2 \leq (1 - \eta_{t+1}^f)^2 \mathbb{E}\|e_t^f\|^2 + 2(\eta_{t+1}^f)^2 \sigma_f^2 + 4(1 - \eta_{t+1}^f)^2 L_K^2 \alpha_t^2 \mathbb{E}\|h_t^f\|^2$$
$$+ 8(1 - \eta_{t+1}^f)^2 L_K^2 \beta_t^2 \mathbb{E}\|e_t^g\|^2 + 8(1 - \eta_{t+1}^f)^2 L_K^2 \beta_t^2 \mathbb{E}\|\nabla_y g(x_t, y_t)\|^2,$$

The proof follows by noticing the fact that for the gradient estimate $h_t^g$ with $\eta_t^g = 1$, we have $\mathbb{E}\|e_t^g\|^2 \leq \sigma_g^2$ from Assumption 4-(ii) and the Lipschitzness of $\nabla_y g(x, \cdot)$ combined with the fact that $\nabla_y g(x, y^*(x)) = 0$. $\qquad\square$

### E.4 Descent in potential function

In this section, we define the potential function as:

$$\widehat{V}_t := (\ell(x_t) - \ell^*) + \|e_t^f\|^2 + \|y_t - y^*(x_t)\|^2, \tag{49}$$

which is different from that of (35). We next show that the potential function decreases with appropriate choice of parameters.

**Lemma E.4.** *With the potential function, $\widehat{V}_t$, defined in (49), with the choice of parameters*

$$\eta_{t+1}^f = (\mu_f + 1)\alpha_t, \ \beta_t = \hat{c}_\beta \alpha_t \ \text{with} \ \hat{c}_\beta = \frac{8L_y^2 + 8L^2 + 2\mu_f}{\mu_g} \ \text{and} \ \gamma_t = \frac{\mu_g \beta_t}{2(1 - \mu_g \beta_t)} \ \text{for all} \ t \in \{0, 1, \ldots, T-1\},$$

*with $\alpha_{-1} = \alpha_0$, moreover, we choose*

$$\alpha_t \leq \left\{ \frac{1}{\mu_f + 1}, \frac{1}{2\mu_g \hat{c}_\beta}, \frac{\mu_g}{\hat{c}_\beta L_g^2}, \frac{1}{8L_K^2 + L_f}, \frac{L^2 + 2L_y^2}{4L_K^2 L_g^2 \hat{c}_\beta^2} \right\}. \tag{50}$$

*Further, we choose*

$$K = \frac{L_g}{2\mu_g} \log \left( \left( \frac{C_{g_{xy}} C_{f_y}}{\mu_g} \right)^2 T \right)$$

*such that we have $\|B_t\|^2 \leq 1/T$, then we have*

$$\mathbb{E}[\widehat{V}_{t+1}] \leq (1 - \mu_f \alpha_{t+1})\mathbb{E}[\widehat{V}_t] + \frac{2\alpha_t}{T} + \left[ (2\hat{c}_\beta^2 + 8\hat{c}_\beta^2 L_K^2)\sigma_g^2 + 2(\mu_f + 1)^2 \sigma_f^2 \right] \alpha_t^2,$$

*for all $t \in \{0, 1, \ldots, T-1\}$.*

*Proof.* From Lemma E.3, we have

$$\mathbb{E}\|e_{t+1}^f\|^2 \leq (1 - \eta_{t+1}^f)\mathbb{E}\|e_t^f\|^2 + 2(\eta_{t+1}^f)^2 \sigma_f^2 + 4L_K^2 \alpha_t^2 \mathbb{E}\|h_t^f\|^2 + 8L_K^2 \beta_t^2 \sigma_g^2 + 8L_K^2 L_g^2 \beta_t^2 \mathbb{E}\|y_t - y^*(x_t)\|^2, \tag{51}$$

which follows from $1 - \eta_{t+1}^f \leq 1$. With the choice of $\eta_t = (\mu_f + 1)\alpha_t$ and $\beta_t = \hat{c}_\beta \alpha_t$ we get from (51):

$$\mathbb{E}\|e_{t+1}^f\|^2 \leq (1 - (\mu_f + 1)\alpha_t)\mathbb{E}\|e_t^f\|^2 + 2(\mu_f + 1)^2\alpha_t^2\sigma_f^2 + 4L_K^2\alpha_t^2\mathbb{E}\|h_t^f\|^2$$
$$+ 8L_K^2\hat{c}_\beta^2\alpha_t^2\sigma_g^2 + 8L_K^2L_g^2\hat{c}_\beta^2\alpha_t^2\mathbb{E}\|y_t - y^*(x_t)\|^2, \tag{52}$$

Next, we consider the descent in the iterates of inner problem. Again using Lemma E.2 we have

$$\mathbb{E}\|y_{t+1} - y^*(x_{t+1})\|^2 \leq (1 + \gamma_t)\left(1 - 2\beta_t\mu_g + \beta_t^2L_g^2\right)\mathbb{E}\|y_t - y^*(x_t)\|^2 \tag{53}$$
$$+ \left(1 + \frac{1}{\gamma_t}\right)L_y^2\alpha_t^2\mathbb{E}\|h_t^f\|^2 + (1 + \gamma_t)\beta_t^2\sigma_g^2.$$

Using the fact that $\beta_t \leq \frac{\mu_g}{L_g^2}$, $\beta_t \leq \frac{1}{2\mu_g}$ and from the choice of $\gamma_t$ we have $1 + \frac{1}{\gamma_t} \leq \frac{2}{\mu_g\beta_t}$ Substituting the $\gamma_t$, $\beta_t$ and the upper bound on $1 + \frac{1}{\gamma_t}$ in (53) above we get:

$$\mathbb{E}\|y_{t+1} - y^*(x_{t+1})\|^2 \leq \left(1 - \frac{\hat{c}_\beta\mu_g\alpha_t}{2}\right)\mathbb{E}\|y_t - y^*(x_t)\|^2 + \frac{2L_y^2\alpha_t}{\mu_g\hat{c}_\beta}\mathbb{E}\|h_t^f\|^2 + 2\hat{c}_\beta^2\alpha_t^2\sigma_g^2. \tag{54}$$

Next, replacing the choice of $\hat{c}_\beta$ in (54), we get:

$$\mathbb{E}\|y_{t+1} - y^*(x_{t+1})\|^2 \leq \left(1 - [4L_y^2 + 4L^2 + \mu_f]\alpha_t\right)\mathbb{E}\|y_t - y^*(x_t)\|^2 + \frac{\alpha_t}{4}\mathbb{E}\|h_t^f\|^2 + 2\hat{c}_\beta^2\alpha_t^2\sigma_g^2. \tag{55}$$

Finally, to construct the potential function defined in (49) we add (52) and (55) to the expression of Lemma E.1, we get

$$\mathbb{E}[\widehat{V}_{t+1}] \leq (1 - \mu_f\alpha_t)\mathbb{E}[\widehat{V}_{t+1}] - \left(\frac{\alpha}{2}(1 - \alpha_tL_f) - \frac{\alpha_t}{4} - 4L_K^2\alpha_t^2\right)\mathbb{E}\|h_t^f\|^2 + 2\alpha_t\|B_t\|^2$$
$$- \left(4L^2\alpha_t + 4L_y^2\alpha_t - 2L^2\alpha_t - 8L_K^2L_g^2\hat{c}_\beta^2\alpha_t^2\right)\mathbb{E}\|y_t - y^*(x_t)\|^2$$
$$+ (2\hat{c}_\beta^2 + 8\hat{c}_\beta^2L_K^2)\alpha_t^2\sigma_g^2 + 2(\mu_f + 1)^2\alpha_t^2\sigma_f^2.$$

Noting the fact that $\alpha_t \leq \frac{1}{8L_K^2 + L_f}$ and $\alpha_t \leq \frac{L^2 + 2L_y^2}{4L_K^2L_g^2\hat{c}_\beta^2}$ and choosing $B_t$ such that we have $\|B_t\|^2 \leq \frac{1}{T}$, we get

$$\mathbb{E}[\widehat{V}_{t+1}] \leq (1 - \mu_f\alpha_t)\mathbb{E}[\widehat{V}_t] + \frac{2\alpha_t}{T} + \left[(2\hat{c}_\beta^2 + 8\hat{c}_\beta^2L_K^2)\sigma_g^2 + 2(\mu_f + 1)^2\sigma_f^2\right]\alpha_t^2.$$

This concludes the proof of the lemma. $\qquad\square$

### E.5   Proof of Theorem 3.3

Next, we conclude the proof for the case of strongly-convex outer objective function case based on fixed step sizes and momentum parameters.

*Proof.* With fixed step sizes, i.e. $\alpha_t = \alpha$ for all $t \in \{0, 1, \dots, T-1\}$, we have from the Lemma E.4

$$\mathbb{E}[\widehat{V}_{t+1}] \leq (1 - \mu_f\alpha)\mathbb{E}[\widehat{V}_t] + \frac{2\alpha}{T} + \left[(2\hat{c}_\beta^2 + 8\hat{c}_\beta^2L_K^2)\sigma_g^2 + 2(\mu_f + 1)^2\sigma_f^2\right]\alpha^2.$$

applying the above inequality recursively we get

$$\mathbb{E}[\widehat{V}_t] \leq (1 - \mu_f\alpha)^t\mathbb{E}[\widehat{V}_0] + \frac{2\alpha}{T}\sum_{k=0}^{t-1}(1 - \mu_f\alpha)^k + \left[(2\hat{c}_\beta^2 + 8\hat{c}_\beta^2L_K^2)\sigma_g^2 + 2(\mu_f + 1)^2\sigma_f^2\right]\alpha^2\sum_{k=0}^{t-1}(1 - \mu_f\alpha)^k$$

$$\overset{(a)}{\leq} (1 - \mu_f\alpha)^t\left((\ell(x_0) - \ell^*) + \mathbb{E}\|e_0^f\|^2 + \mathbb{E}\|y_0 - y^*(x_0)\|^2\right) + \frac{2\alpha}{T}\sum_{k=0}^{t-1}(1 - \mu_f\alpha)^k$$

$$+ \left[(2\hat{c}_\beta^2 + 8\hat{c}_\beta^2L_K^2)\sigma_g^2 + 2(\mu_f + 1)^2\sigma_f^2\right]\alpha^2\sum_{k=0}^{t-1}(1 - \mu_f\alpha)^k$$

$$\overset{(b)}{\leq} (1 - \mu_f\alpha)^t\left\{(\ell(x_0) - \ell^*) + \sigma_f^2 + \|y_0 - y^*(x_0)\|^2\right\} + \frac{2}{\mu_fT} + \frac{(2\hat{c}_\beta^2 + 8\hat{c}_\beta^2L_K^2)\sigma_g^2 + 2(\mu_f + 1)^2\sigma_f^2}{\mu_f}\alpha, \tag{56}$$

where $(a)$ follows from the definition of $\widehat{V}_t$ given in (49) and $(b)$ utilizes the summation of a geometric progression.

This concludes the proof of the theorem. $\qquad\square$

**Sample complexity of SUSTAIN in the strongly convex setting** Let us estimate the total number of iterations, $T$, needed to reach an $\epsilon$-optimal solution. First, we select a constant step size such that

$$\alpha \leq \frac{\mu_f}{4\big[(2\hat{c}_\beta^2 + 8\hat{c}_\beta^2 L_K^2)\sigma_g^2 + 2(\mu_f + 1)^2\sigma_f^2\big]}\epsilon \quad \Longrightarrow \quad \frac{\big[(2\hat{c}_\beta^2 + 8\hat{c}_\beta^2 L_K^2)\sigma_g^2 + 2(\mu_f + 1)^2\sigma_f^2\big]}{\mu_f}\alpha \leq \frac{\epsilon}{4}, \tag{57}$$

which controls the last term in (56). Secondly, to control the second term in (56), we observe that $T \geq \frac{8}{\mu_f\epsilon}$ implies $\frac{2}{\mu_f T} \leq \frac{\epsilon}{4}$. Finally, controlling the first term in (56) requires

$$\frac{\epsilon}{2} \geq (1 - \mu_f\alpha)^T\big((\ell(x_0) - \ell^*) + \sigma_f^2 + \|y_0 - y^*(x_0)\|^2\big) \tag{58}$$

which means we require:

$$(1 - \mu_f\alpha)^T \leq \frac{\epsilon}{2\big((\ell(x_0) - \ell^*) + \sigma_f^2 + \|y_0 - y^*(x_0)\|^2\big)}$$

$$\Longleftrightarrow T\log(1 - \mu_f\alpha) \leq \log\left(\frac{\epsilon}{2\big((\ell(x_0) - \ell^*) + \sigma_f^2 + \|y_0 - y^*(x_0)\|^2\big)}\right)$$

$$\Longleftrightarrow T \geq \frac{\log\left(\frac{2\big((\ell(x_0)-\ell^*)+\sigma_f^2+\|y_0-y^*(x_0)\|^2\big)}{\epsilon}\right)}{-\log(1 - \mu_f\alpha)}$$

$$\overset{(a)}{\Longleftarrow} T \geq \log\left(\frac{2\big((\ell(x_0) - \ell^*) + \sigma_f^2 + \|y_0 - y^*(x_0)\|^2\big)}{\epsilon}\right)\frac{1}{\mu_f\alpha}, \tag{59}$$

where $(a)$ is due to $\log x \leq x - 1$ for all $x > 0$. This along with (57) imply that we require at most $T = \tilde{\mathcal{O}}(\epsilon^{-1})$ iterations to reach an $\epsilon$-optimal solution, i.e., $\mathbb{E}[\ell(x_t) - \ell^*] \leq \epsilon$. Finally, as each iteration takes a batch of $K = \mathcal{O}(\log(T))$ samples, the total sample complexity required to reach an $\epsilon$-optimal solution is bounded as $T = \tilde{\mathcal{O}}(\epsilon^{-1})$. $\qquad\square$