# OpenReview forum: "A Near-Optimal Algorithm for Stochastic Bilevel Optimization via Double-Momentum"
_NeurIPS.cc/2021/Conference — NeurIPS 2021 Poster_

### Official Review · Reviewer_wA7a · 2021-07-11

**Rating:** 7
**Confidence:** 3

**Summary:**

The work developed a new algorithm for unconstrained bilevel optimization with strongly convex lower level subproblems. The proposed algorithm executes on a single-timescale, without the need to use either two-timescale updates, large batch gradients, or double-loop algorithm. Authors showed that SUSTAIN is both sample and computation efficient, because it matches the best-known sample complexity guarantees on single-level problems with non-convex and strongly convex objective functions, while matching the best-known per-iteration computational complexity for the same class of bi-level problems.

**Limitations And Societal Impact:**

The limitations have been addressed.

**Main Review:**

The paper is well written and the demonstration is clear.

**Time Spent Reviewing:**

1 hour

---

> ### Author Response · Authors · 2021-08-10
> **Response to review**
>
> We greatly thank the reviewer for the positive evaluation of our work.

---

### Official Review · Reviewer_LvzC · 2021-07-15

**Rating:** 7
**Confidence:** 4

**Summary:**

This paper proposes an algorithm with double momentum solve stochastic bilevel optimization. Both the sample complexity and per-iteration complexity are state-of-the-art.

**Limitations And Societal Impact:**

Yes.

**Main Review:**

Merits: optimal sample complexity, low per-iteration complexity by replacing Hessian inverse process by an estimator from reference [14].

Flaws:
1. Some extra assumptions are made under Lemma C.1 in the appendix, including bounded stochastic gradients. It may help to make the settings more clear by moving these assumption to the main body of the paper. These are critical assumption in the area.

2. The authors cite Lemma C.1 from reference [18] without providing a proof here. I could not figure out how eq(106) is proved in [18] (or corresponding terms in [14]).

**Time Spent Reviewing:**

3

---

> ### Author Response · Authors · 2021-08-10
> **Clarification of Lemma C1**
>
> We thank the reviewer for the comments and suggestions. Below we address the comments in detail.
>
> > **Your comment 1:** Some extra assumptions are made under Lemma C.1 in the appendix, including bounded stochastic gradients. It may help to make the settings more clear by moving these assumptions to the main body of the paper. These are critical assumptions in the area.
>
> **Our response:** We thank the reviewer for the suggestion. We will add the assumptions of Lemma C1 in the main body of the paper. Next, we describe the proof of equation (106) from [18].
>
>
> > **Your comment 2:** he authors cite Lemma C.1 from reference [18] without providing a proof here. I could not figure out how eq(106) is proved in [18] (or corresponding terms in [14]).
>
> **Our response:** From the authors' understanding, Equation (106) in [18] can be derived from the strong convexity of the lower-level problem. More specifically, we observe that $H_{yy}$ in [18] is defined as:
> $$
> H_{yy} = \frac{K}{L_g}  \prod_{i  = 1}^{k(K)}    \bigg( I_{d_\ell} - \frac{\nabla_{yy}^2 g(x, y; \zeta^{(i)})}{L_g} \bigg)
> $$
> From the Lipschitz smoothness and strong-convexity of $g(x, y ; \zeta^{(i)})$ we have:
> $$
> \mu_g I_{d_\ell} \preceq  \nabla_{yy}^2 g(x, y; \zeta^{(i)}) \preceq L_g I_{d_\ell},
> $$
> This implies that:
> $$
>   I_{d_\ell} - \frac{\nabla_{yy}^2 g(x, y; \zeta^{(i)})}{L_g}  \preceq \bigg( 1 - \frac{\mu_g}{L_g}  \bigg)  I_{d_\ell}  \quad (1)
> $$
> Then to bound $\mathbb{E}[\||H_{yy}\||]$, we utilize the following:
> $$
>    \mathbb{E}[\||H_{yy}\||]  \overset{(a)}{\leq} \frac{K}{L_g}  \mathbb{E} \bigg[ \bigg( 1 - \frac{\mu_g}{L_g} \bigg)^{k(K)}  \bigg] \overset{(b)}{=} \frac{1}{L_g} \sum_{k = 0}^{K-1} \bigg( 1 - \frac{\mu_g}{L_g}\bigg)^k \overset{(c)}{\leq} \frac{1}{\mu_g},
> $$
> where inequality $(a)$ follows from (1) the fact that $I_{d_\ell} - \frac{\nabla_{yy}^2 g(x, y; \zeta^{(i)})}{L_g}$ are independent and identically distributed conditioned on $k(K)$; equality $(b)$ results from taking expectation with respect to $k(K)$ which is chosen uniformly at random from the set ${0, \ldots, K-1}$; finally, inequality $(c)$ utilizes the summation of a geometric series.

---

> > ### Comment · Reviewer_LvzC · 2021-08-21
> > **Thank you for your response.**
> >
> > Overall, I am happy to keep my positive rating of 7. Glad that you decide to move the assumptions to main body. Also, please note that in your response proving (106) in [18], you use bounded assumption of stochastic second order gradient of g, which is another critical assumption that should be made clear in the paper. Good luck.

---

> > > ### Author Response · Authors · 2021-08-25
> > > **Regarding Assumptions**
> > >
> > > We thank the reviewer for valuable suggestions and careful review of our paper. We note that the bounded assumption of the stochastic second-order gradient of $g$ is implied by Assumption 3 stated in Lines 141-142 of the original paper. We will make this clear in the revised version of the paper.

---

> > > > ### Comment · Reviewer_LvzC · 2021-08-31
> > > > **Just a conversation about assumptions in the literature.**
> > > >
> > > > Thank you for pointing out the bounded assumption of the stochastic second-order gradient of
> > > > $g$ in your paper. By the way, I don't find that assumption explicitly made in [14, 18]. I think those authors missed something.
> > > >
> > > > (NOTE: This comment has nothing  to do with evaluation of this submission.)

---

> > > > > ### Author Response · Authors · 2021-08-31
> > > > > **Response: About assumptions in the literature**
> > > > >
> > > > > The reviewer is correct that the **bounded assumption of the stochastic second-order gradient** of $g$ is not explicitly stated as a separate assumption in [14] and [18], however, it is mentioned in the second paragraph after Assumption 3 (on Page 3) of [14]. Precisely, the authors state that they make the same assumptions for the stochastic functions as for the deterministic functions, which implies boundedness of the stochastic gradients of function $g$.

---

### Official Review · Reviewer_YpAh · 2021-07-18

**Rating:** 6
**Confidence:** 4

**Summary:**

This paper tackles stochastic unconstrained bilevel optimization problems, where the lower level subproblem is strongly-convex and the upper level objective function is smooth. By applying momentum-assisted gradient estimator for both the upper and lower level updates, the paper proposes a new bilevel stochastic optimization method termed SUSTAIN. It has been established that the sample complexity of SUSTAIN matches the best-known complexity for single-level stochastic gradient algorithms.

**Limitations And Societal Impact:**

The authors adequately addressed the limitations and potential negative societal impact of their work.

**Main Review:**

The paper is generally well-written. The followings are the positive points of the paper.
+	By applying momentum-assisted gradient estimator for both the upper and lower updates, the paper develops a new momentum-based bilevel optimization method.
+	The paper established that the new stochastic bilevel optimization method matches the best-known complexity for single-level stochastic gradient algorithms.
+	Some simulations have shown the empirical advantage of the proposed algorithm.

However, I have concerns about the paper in the following aspects.
-	The paper claims that it is a single-loop algorithm. However, by looking into the implementation of SUSTAIN, it seems that it needs another loop to calculate (7).
-	The authors pose an interesting question at the bottom of page 5 that ``It is not clear which are the most important objects for which variance reduction shall be applied.’’ It seems that this question has not been addressed explicitly in the paper.
-	The reduction of the SUSTAIN theory to existing results needs further elaboration. Specifically, what is the convergence rate if the momentum of lower-level update is off, e.g., eta^g=1; what is the convergence rate if the momentum of upper-level update is off, e.g., eta^f=1. Authors do provide some discussion in the strongly convex setting, but this should also be discussed in the nonconvex setting.
-	The simulation is not convincing. The curve HOAG in Fig 1 looks a bit strange. It seems that authors do not plot all the accuracy at all the iterations but only at 2-3 iterations. It is important to see the performance of HOAG at the initial stage. Any particular reason for this?

I am happy to raise my score if my concerns are addressed appropriately.


========= After rebuttal =======

Thanks for answering my questions. Most of my initial concerns have been addressed. I encourage the authors to add the discussion.
I updated my score accordingly.


**Time Spent Reviewing:**

5

---

> ### Author Response · Authors · 2021-08-10
> **Impact of upper and lower level momentum terms**
>
> We thank the reviewer for the comments. Next, we address the comments in detail.
>
> > **Your comment 1:** The paper claims that it is a single-loop algorithm. However, by looking into the implementation of SUSTAIN, it seems that it needs another loop to calculate (7).
>
> **Our response:** We would like to clarify that the upper-level stochastic gradient estimate in (7) is not computed by using an iterative process (i.e., a loop). The estimator in (7) is computed via a small batch (of the order of $\log (\epsilon^{-1})$) of lower-level functions, where $\epsilon$ is the desired solution accuracy (or $\epsilon$-stationarity).  For example, in the experiments, we have used a small sample size of $K = 10$, to compute the upper-level gradient estimate (7). This computation is one-shot, as long as we have $\log(\epsilon^{-1})$ samples available. When we say this algorithm is a single-loop algorithm, we meant that we only need to update one step (using a minibatch of samples) for the inner problem, followed by one step (using a minibatch of samples) for the outer problem. We do not need to run multiple steps of inner problems, as opposed to a number of recent works in the literature.
>
> > **Your comment 2:** The authors pose an interesting question at the bottom of page 5 that ``It is not clear which are the most important objects for which variance reduction shall be applied.’’ It seems that this question has not been addressed explicitly in the paper.
> The reduction of the SUSTAIN theory to existing results needs further elaboration. Specifically, what is the convergence rate if the momentum of lower-level update is off, e.g., $\eta^g=1$; what is the convergence rate if the momentum of upper-level update is off, e.g., $\eta^f=1$. Authors do provide some discussion in the strongly convex setting, but this should also be discussed in the nonconvex setting.
>
> **Our response:** We address the question about the effect of momentum for the two cases in the non-convex setting.
>
> **[Case I: $\eta_t^g \equiv 1$]**  Below we show that when SUSTAIN utilizes only the upper-level momentum without lower-level momentum, it improves the convergence of TTSA and achieves the same convergence as vanilla single-level SGD algorithm. Specifically, we have the following result.
>
> **Theorem:** Under the same set of assumptions as in Theorem 3.2 and with $K = ( {L_g} / {2\mu_g} ) \log ( ( {C_{g_{xy}} C_{f_y}} / {\mu_g} )^2 T^{1/2} )$ Set the parameters $\alpha_t \equiv \alpha$, $\eta_{t}^f \equiv \alpha$, $\beta_{t} \equiv c_\beta \alpha$ with $c_\beta = (16 L^2 L_y^2 + 1) / \mu_g$, $\eta_t^g \equiv 1$ and with  $\alpha = \mathcal{O}(1/\sqrt{T})$, we have:
> $$
> \mathbb{E} [ \|| \nabla \ell(x_{a(T)}) \||^2 ] \leq \Delta_0 \alpha + 4 / \sqrt{T},
> $$
> where we have defined  $\Delta_0 : = (\ell(x_0) - \ell^\ast) + L^2 \|y_0 - y^\ast(x_0)\|^2 + 5 \sigma_f^2 + 8 (L_K^2 + L^2) c_\beta^2 \sigma_g^2$.
>
> The above result implies that $\mathbb{E} [ \|| \nabla \ell(x_{a(T)}) \||^2 ] = {\cal O}(1/\sqrt{T})$.  This further implies that to achieve an $\epsilon$-stationary solution SUSTAIN with only upper-level momentum requires $\mathcal{O}(\epsilon^{-2})$ stochastic samples for both upper and lower-level functions. Note that this improves over TTSA which utilizes a vanilla SGD update for both upper and lower-level problems, i.e., $\eta_t^f \equiv 1$ and  $\eta_t^g \equiv 1$ and requires $\mathcal{O}(\epsilon^{-2.5})$ stochastic samples for both upper and lower-level functions.
>
> **[Case II: $\eta_t^f \equiv 1$]** For the case when the upper-level momentum is turned-off, i.e. with $\eta_t^f \equiv 1$, SUSTAIN does not improve upon the performance of TTSA. Note that for this case SUSTAIN requires $\mathcal{O}(\epsilon^{-2.5})$ stochastic samples to achieve an $\epsilon$-stationary solution which is same as TTSA.
>
> To address the reviewer's concern in the revised version of the manuscript we will add additional discussions on the effect of the momentum parameter in the non-convex setting.
>
> > **Your comment 3:** The simulation is not convincing. The curve HOAG in Fig 1 looks a bit strange. It seems that authors do not plot all the accuracy at all the iterations but only at 2-3 iterations. It is important to see the performance of HOAG at the initial stage. Any particular reason for this?
>
> **Our response:** In Figure 1, we compare the performance of different algorithms with the number of overall gradient computations for a fair comparison.  Since HOAG is a *deterministic* algorithm, within each iteration it utilizes all the samples to compute the gradient. This implies that with a total of $3 \times 10^4$ gradient evaluations HOAG updates the parameters only 3 to 4 times, therefore, its performance can only be evaluated after each update. This is the reason that there are only 2-3 points for HOAG in Fig. 1. In contrast, SUSTAIN and stocBio are *stochastic* algorithms, so they can utilize smaller batch-size gradients and perform multiple parameter updates with the same number of overall gradient evaluations. In the updated version of the paper, we will clarify this point in more detail. Moreover, in the Appendix, we will evaluate the algorithms' performance using a larger threshold of total gradient evaluation, so that HOAG can run for more iterations.

---

### Official Review · Reviewer_dpaA · 2021-07-20

**Rating:** 7
**Confidence:** 3

**Summary:**

This paper investigates a bi-level stochastic optimization problem in which the upper level function depends on the minimizer of the lower level objective. A new algorithm based on stochastic momentum-assisted gradient estimator is introduced, by exploiting prior iterations gradient estimates to improve the quality of the current gradient estimation (by reducing its variance). The new algorithm matches the best complexity bounds as the optimal SGD algorithms for single-level optimization.

**Ethics Review Area:**

["I don’t know"]

**Limitations And Societal Impact:**

The paper is theoretical in nature and the reviewer does not see any evident negative societal impact.

**Main Review:**

Overall comments

Table 1 provides an excellent and clear overview of existing results in bi-level optimization problems and a clear position of the new contributions; much appreciated. The paper is concise, well written and the theoretical results seem correct and improve upon the existing state of the art.


Detailed comments

1. Does the Stackelberg equilibrium in non-cooperative game theory fall in this class of bi-level optimization class? If so, this also could be mentioned in the introduction section.

2. The curves in the figures of the numerical results are not distinguishable (in black and white print). Some markers could be included for clarity.

3. Theorem 3.3.  does not include the low-level momentum term. Some explanations on why this term is not required in strongly convex problems should be provided.

4. The remark in the conclusion section regarding the rigurous lower bound for the sample complexity is not very clear. Some more details should be provided here.



**Time Spent Reviewing:**

2h

---

> ### Author Response · Authors · 2021-08-10
> **Importance of momentum term**
>
> We thank the reviewer for the positive comments. Below, we provide our response to the specific comments.
>
> > **Your comment 1:** Does the Stackelberg equilibrium in non-cooperative game theory fall in this class of bi-level optimization class? If so, this also could be mentioned in the introduction section.
>
> **Our response:** The bi-level problem we address can be used to model the Stackelberg game. Below we provide an example of Stackelberg competition that arises in the case of two firms. Suppose $P_l$ denotes the profit earned by the leader firm and $P_f$ the profit earned by the follower firm. Then the Stackelberg game between the two firms is
>
> $$
> \max_x P_l(x, y^\ast(x)) ~~\text{s.t.} ~~y^\ast(x) \in \arg \max_y P_f(x, y).
> $$
> Note that this is a special case of the stochastic bilevel problem considered in the paper (see eq (1) in the paper) with $f(x,y) = - P_{l}(x , y)$ and $g(x,y) = -P_f(x,y)$. In the revised version of the manuscript, we will update the introduction with the discussion of Stackelberg games.
>
> > **Your comment 2:** The curves in the figures of the numerical results are not distinguishable (in black and white print). Some markers could be included for clarity.
>
> **Our response:** To distinguish different plots in the figures, we will include plots with markers.
>
>
> > **Your comment 3:** Theorem 3.3. does not include the low-level momentum term. Some explanations on why this term is not required in strongly convex problems should be provided.
>
> **Our response:** For strongly convex problems, the iterates generated by the $x$-update introduce less variance into the lower-level problem compared to the non-convex case. Therefore, for strongly convex problems momentum is not required, because the momentum is utilized to reduce the variance of the gradient estimators. A straightforward way to see such an intuition is as follows: it is known that for a  single-level strongly convex stochastic problem, Momentum-SGD does not provide improvement in the convergence rate compared to vanilla SGD. In the revised version of the manuscript, we will add the above discussion.
>
> > **Your comment 4:** The remark in the conclusion section regarding the rigorous lower bound for the sample complexity is not very clear. Some more details should be provided here.
>
> **Our response:** For the stochastic bilevel problems of the form (1) considered in the paper, the lower bounds which establish the optimality of proposed algorithms are not yet known. Therefore, in the future, we plan to derive these lower bounds and show that the SUSTAIN algorithm is in fact optimal under the set of assumptions made in the paper. In the revised version of the paper, we will clarify the remark regarding the lower bounds in the conclusion section.

---

> > ### Comment · Reviewer_dpaA · 2021-08-22
> > **Authors' replies**
> >
> > The reviewer is happy with the authors' replies and therefore the initial assessment 7 stands.

---

### Decision · Program_Chairs · 2021-09-27

**Decision:**

Accept (Poster)

**Comment:**

Large-scale bi-level and multi-level optimization has several emerging applications (e.g., robotics) where optimizers are themselves embedded in end-to-end deep learning pipelines as layers. Bilevel problems where the lower level subproblem is strongly-convex and the upper level objective function is smooth is an interesting special case amenable to complexity bound analyses for reaching a stationary point. As such, the contributions of this paper should interest both numerical optimization and application communities. The reviews ask for a few clarifications in the final revision.